# A Geometry-Aware Metric for Mode Collapse in Time Series Generative Models

**Yassine Abbahaddou**[*]
LIX, École Polytechnique
Institut Polytechnique de Paris
yassine.abbahaddou@polytechnique.edu

**Amine Mohamed Aboussalah**[*]
Department of Finance and Risk Engineering
Tandon School of Engineering
New York University
ama10288@nyu.edu

## Abstract

Generative models such as Generative Adversarial Networks (GANs), Variational Autoencoders (VAEs), and diffusion models often suffer from *mode collapse*, failing to reproduce the full diversity of their training data. While this problem has been extensively studied in image generation, it remains largely unaddressed for time series. We introduce a formal definition of mode collapse for time series and propose DMD-GEN, a geometry-aware metric that quantifies its severity. DMD-GEN leverages *Dynamic Mode Decomposition* (DMD) to extract coherent temporal structures and uses *Optimal Transport* between DMD eigenvectors to measure discrepancies in underlying dynamics. By representing the subspaces spanned by the DMD eigenvectors as point structures on a Grassmann manifold, and comparing them via Wasserstein distances computed from principal angles, DMD-GEN enables a principled geometric comparison between real and generated sequences. The metric is efficient, requiring no additional training, supports mini-batch evaluation, and is easily parallelizable. Beyond quantification, DMD-GEN offers interpretability by revealing which dynamical modes are distorted or missing in the generated data. Experiments on synthetic and real-world datasets using TimeGAN, TimeVAE, and DiffusionTS show that DMD-GEN aligns with existing metrics while providing the first principled framework for detecting and interpreting mode collapse in time series. Our code is available at: here.

## 1 Introduction

Generative models have gained significant attention in recent years, driven by recent advancements in computational power, the availability of extensive datasets, and breakthrough developments in machine learning algorithms. Notably, models like GANs and VAEs excel at capturing rich and meaningful latent representations of data [17, 23, 38, 41, 45]. These models are applied in various ways, such as generating realistic samples that mimic real-world data distributions [51], modeling complex probability distributions through density estimation [59], and augmenting datasets with synthetic data to improve model generalization [25, 61], among others. However, recent studies have revealed that generative models sometimes fail to produce diverse samples, leading to reduced effectiveness in applications that require a broad spectrum of variations [1, 6, 40]. An illustration of this challenge can be seen in GANs, which often experience mode collapse, a phenomenon where the generator focuses on a limited subset of the data distribution, leading to the production of repetitive or similar samples rather than capturing the full diversity of the training data [3, 4, 15, 42]. VAEs also face a phenomenon called posterior collapse, where the model tends to generate outputs that are similar or indistinguishable for different inputs. This limitation reduces the model's ability to produce

---

[*]Equal contribution.

39th Conference on Neural Information Processing Systems (NeurIPS 2025).

diverse samples [18, 56]. Diffusion models, while generally robust against mode collapse compared to GANs and VAEs, are not entirely immune to difficulties in covering the full data distribution. These challenges often emerge under strong classifier-free guidance or limited data regimes [20, 46, 50].

The issue of diversity in generative models has received significant attention in fields such as computer vision and natural language processing [11, 21, 29, 32, 49], however, it remains relatively underexplored for time series data. The inherently time-dependent and dynamic nature of time series makes traditional definitions of mode collapse insufficient, highlighting the need for a framework tailored to time-dependent data. Defining modes in time series is particularly challenging, as it requires capturing evolving temporal patterns rather than simply avoiding repetitive or static sequences. A natural way to determine if a generative model preserves the diversity is to evaluate the similarity between original and generated time series. However, widely used existing evaluation metrics, such as, Predictive and Discriminative Scores [60], and Contextual FID [22], suffer from key limitations. They are often computationally expensive, since they rely on training auxiliary models that capture temporal dependencies in the data. More importantly, these metrics provide only aggregate performance indicators and fail to reveal which dynamic modes have been preserved or lost in the generated sequences.

To address these challenges, we introduce a time-series-specific definition of a mode. Our approach is based on Dynamic Mode Decomposition (DMD) [52], a spectral method that identifies dominant coherent structures in temporal dynamics. This allows to develop a new metric that is interpretable, training-free, and explicitly quantifies the preservation of temporal modes in generative models. The main contributions of this work are as follows,

- **New Definition of Mode Collapse for Time Series:** We introduce a new definition of mode collapse specifically for time series data, leveraging DMD to capture and analyze coherent dynamic patterns.

- **Development of DMD-GEN Metric:** We propose DMD-GEN, a new metric to detect mode collapse, which consistently aligns with traditional generative model evaluation metrics while offering unique insights into time series dynamics.

- **Enhanced Interpretability:** The DMD-GEN metric provides increased interpretability by decomposing the underlying dynamics into distinct modes, allowing for a clearer understanding of the preservation of essential time series characteristics.

- **Efficiency in Time Complexity:** Our approach offers significant computational efficiency as it requires no additional training, making it highly scalable for real-time applications.

## 2   Background and Related Work

**Mode Collapse for Time Series.**   Before delving into the details of our new evaluation metric that incorporates the concepts mentioned above, it is worth highlighting the challenges to be addressed in order to measure mode collapse when dealing with time series. *(i) Capturing Modes:* For time series, we need to consider *modes* that represent different evolving patterns over time. Real-world time series data rarely exhibit a single clean pattern [26, 30]. Instead, time series data often exhibit multiple patterns simultaneously, e.g. layered over long-term trends and shorter-term fluctuations representing evolving modes. This makes it difficult to isolate and identify the specific mode of interest. Moreover, in contrast to data modalities with well-defined discrete structures, such as images or text, time series data exhibit inherent temporal continuity. This makes it challenging to determine the beginning and end of a specific evolving pattern (mode). *(ii) Similarity Measurement:* Time series data often have different characteristics that make standard distance metrics such as Euclidean distance less effective as a similarity measure. For instance, Euclidean distance is sensitive to variations in feature scales across dimensions. In high-dimensional spaces, this can cause a few dimensions to dominate, leading to distorted similarity measurements. In addition, Euclidean distance does not take into account the temporal dependencies present in the time series data. It assumes that each timestamp is independent, which is inappropriate for time series data where the ordering and temporal dependencies between observations are essential. More advanced similarity measures, such as Dynamic Time Warping (DTW) [37], address temporal dependencies by aligning sequences to minimize distance. However, if fails to capture the underlying modes or coherent dynamic patterns present in the data. This inability to recognize and preserve the essential modes means DTW falls short in assessing mode collapse.

Unlike in images, the mode collapse issue in time series cannot be easily distinguished with human eyes. Therefore, this area remains relatively under-explored. Few studies have formally addressed this problem and proposed solutions within the time series domain. [31] introduced an auto-normalization heuristic that normalizes each time series separately rather than the dataset as a whole. However, the custom auto-normalization heuristic addresses only mode collapse resulting from offset differences between time series, without considering whether the model captures the global trends and seasonality of the dataset. Additionally, *DC-GAN* [36] is the first time series GAN capable of generating all temporal features within a multimodal distributed time series. Based on directed chain stochastic differential equations (DC-SDEs) [13], the model introduces an approach to temporal generation. Although the authors did not explicitly discuss mode collapse, it would be valuable to investigate whether DC-GAN successfully captures the full range of trends and seasonal patterns present in the dataset.

**Dynamic Mode Decomposition (DMD).** *DMD* [43, 52] is a data-driven and model-free method used for analyzing the underlying dynamics of complex systems such as fluid dynamics. It is used to extract modal representations of a nonlinear dynamical system directly from data, without requiring prior knowledge of the system. Given a dynamical system $\dot{x}(t) = \mathbf{f}(\mathbf{x}(t), t; \mu)$, where $\mathbf{x}(t) \in \mathbb{R}^n$ denotes the $n$-dimensional state vector, $\mu \in \mathbb{R}^p$ represents the system parameters, and $\mathbf{f} : \mathbb{R}^n \times \mathbb{R} \times \mathbb{R}^p \to \mathbb{R}^n$ defines the dynamics, DMD approximates the system locally by $\dot{x} \approx \mathcal{A}\mathbf{x}$, where $\mathcal{A}$ is the best-fit linear operator obtained via regression to approximate $\mathbf{f}$. This linear approximation allows for the representation of the system's behavior in a simplified framework and helps construct reduced-order models that capture the essential dynamics of the systems. This is particularly useful for systems with large state spaces, such as fluid dynamics [24, 28]. Analogously, we consider a discrete-time approximation of the dynamical system. Given $\mathbf{x} : t \in \mathbb{R} \mapsto \mathbf{x}(t) \in \mathbb{R}^n$, we collect $m$ consecutive snapshots to construct two data matrices $\mathbf{X}, \mathbf{X}' \in \mathbb{R}^{n \times m}$ defined as

$$\mathbf{X} = \begin{bmatrix} | & | & & | \\ \mathbf{x}_0 & \mathbf{x}_1 & \cdots & \mathbf{x}_{m-1} \\ | & | & & | \end{bmatrix}, \quad \mathbf{X}' = \begin{bmatrix} | & | & & | \\ \mathbf{x}_1 & \mathbf{x}_2 & \cdots & \mathbf{x}_m \\ | & | & & | \end{bmatrix}.$$

These snapshots are taken with a time-step $\Delta t$ small enough to capture the highest frequencies in the system's dynamics, i.e., $\forall k \in \mathbb{N}$, $\mathbf{x}_k = \mathbf{x}(k\Delta t)$. Assuming uniform sampling in time, we approximate the dynamical system linearly as $\mathbf{x}_{k+1} \approx \mathbf{A}^\star \mathbf{x}_k$, where $\mathbf{A}^\star \in \mathbb{R}^{n \times n}$ is the best-fit operator, i.e., $\mathbf{A}^\star = \arg\min_{\mathbf{A}} \|\mathbf{X}' - \mathbf{A}\mathbf{X}\|_F = \mathbf{X}'\mathbf{X}^\dagger$, where $\|.\|_F$ is the Frobenius norm and $\mathbf{X}^\dagger$ is the Moore-Penrose generalized inverse of $\mathbf{X}$. The optimal discrete-time operator $\mathbf{A}^\star$ is related to the continuous-time operator $\mathcal{A}$, defined earlier, through the exponential mapping $\mathbf{A}^\star = \exp(\mathcal{A}\,\Delta t)$; see Appendix A. The DMD operator $\mathbf{A}^\star$ is deeply rooted in Koopman operator theory [9, 33, 39, 47], which provides a linear perspective on nonlinear dynamical systems. Specifically, DMD can be viewed as a finite-dimensional approximation of the infinite-dimensional Koopman operator that advances observables of the system forward in time. This connection, originally established by [48], enables spectral analysis of nonlinear dynamics through linear algebraic techniques. In essence, Koopman theory offers a principled framework for globally linearizing nonlinear dynamics, allowing DMD to capture coherent spatiotemporal structures and their evolution through the system's spectrum [12, 34, 35]. By analyzing the eigenvalues $\mathbf{\Lambda} = \mathrm{diag}(\lambda_1, \ldots, \lambda_r)$ and the corresponding eigenvectors $\mathbf{\Phi} = [\phi_1, \ldots, \phi_r] \in \mathbb{R}^{n \times r}$ of the DMD operator $\mathbf{A}^\star$, where $r$ denotes the rank of $\mathbf{X}$, we can capture the dominant dynamic patterns that govern the system's evolution. These spectral components, the eigenvalues encoding temporal behavior and the eigenvectors capturing spatial coherence, together provide a compact yet expressive representation of the underlying dynamics. Conventionally, the eigenvectors and their corresponding eigenvalues are arranged in descending order based on the magnitude of the eigenvalues, i.e., $|\lambda_1| \geq \ldots \geq |\lambda_r|$.

Throughout this paper, we distinguish between two related quantities: $r$ denotes the full numerical rank of the snapshot matrix $\mathbf{X}$, corresponding to the total number of available DMD modes obtained from the spectral decomposition of $\mathbf{A}^\star$. In contrast, $k \leq r$ denotes the number of dominant modes retained for reduced-order approximation or comparison. Hence, $r$ characterizes the complete spectral dimension of the system, while $k$ represents the truncated subspace capturing the most significant dynamics. This distinction will be maintained consistently in subsequent sections, where all definitions and metrics involving $\mathcal{M}_k(\mathbf{X})$ refer to the $k$ leading DMD modes.

For a high-dimensional state vector $\mathbf{x} \in \mathbb{R}^n$, the matrix $\mathbf{A}^\star$ comprises $n^2$ elements, making its representation and spectral decomposition computationally challenging. To address this, we apply

dimensionality reduction to efficiently compute the dominant eigenvalues and eigenvectors of $\mathbf{A}^\star$ by constructing a reduced-order approximation $\tilde{\mathbf{A}} \in \mathbb{R}^{r \times r}$. The DMD approximation at each time step can be expressed as follows,

$$\forall t, \quad \mathbf{x_t} = \sum_{j=1}^{r} \phi_j \lambda_j^t b_j = \boldsymbol{\Phi} \boldsymbol{\Lambda}^\mathbf{t} \mathbf{b}, \tag{1}$$

where $\phi_j$ are the DMD modes (eigenvectors of the matrix $\mathbf{A}$), $\lambda_j$ are the corresponding DMD eigenvalues, and $b_j$ denotes the mode amplitude, given by $\mathbf{b} = \boldsymbol{\Phi}^\dagger \mathbf{x}_0$ in matrix form. The detailed steps for this process are provided in the Appendix A.3. DMD modes can be interpreted as basis vectors spanning a subspace that captures coherent spatiotemporal patterns among the components of $\mathbf{x}(t)$. It decomposes a complex time series into a collection of simpler, coherent modes, where each of them captures a specific aspect of the system's behavior, e.g., an oscillation, an exponential growth/decay, or a traveling wave [58]. The DMD modes are spatial fields that often reveal coherent structures within the flow. These structures are fully characterized by the DMD eigenvalues $\boldsymbol{\Lambda}$ and eigenvectors $\boldsymbol{\Phi}$, which respectively encode the temporal frequencies and spatial patterns of the underlying dynamics. Specifically, the imaginary part of the eigenvalues $\boldsymbol{\Lambda}$ determines the oscillation frequency, while the real part indicates the rate of exponential growth or decay [10, 54]. The corresponding DMD eigenvectors $\boldsymbol{\Phi}$ capture the spatial coherence associated with each mode, providing an interpretable link between the system's temporal evolution and its spatial distribution.

## 3 Quantifying Mode Collapse in Time Series

**Notations.** We denote by $\mathcal{G}$ a generative model specific to time series. This model is trained on a dataset comprising $N$ time series, each with a fixed length, represented as $\{\mathbf{X}_i\}_{i=1}^{N}$. During the inference phase, we synthetically generate a set of $\widetilde{N}$ time series, denoted as $\{\widetilde{\mathbf{X}}_j\}_{j=1}^{\widetilde{N}}$ using $\mathcal{G}$. Both the original and generated time series are assumed to have a consistent length, denoted as $\ell$, and dimensionality, represented as $n$. Formally, for any pair of indices $i$ and $j$, the original and generated time series $\mathbf{X}_i$ and $\widetilde{\mathbf{X}}_j$ are elements of the Euclidean space $\mathbb{R}^{n \times \ell}$.

### 3.1 Defining Temporal Modes

In this section, we formalize the notion of temporal modes. Definition 3.1 captures the essence of a mode in terms of the dominant eigenvalues and eigenvectors of the DMD operator, highlighting the significant dynamic structures in the time series data.

**Definition 3.1 (Temporal Modes).** Given a time series $\mathbf{X} = [\mathbf{x}_1, \ldots, \mathbf{x}_\ell] \in \mathbb{R}^{\ell \times n}$, we define the set of temporal modes $\mathcal{M}_k(\mathbf{X})$ as the $k$ dominant eigenvectors $\{\phi_1, \ldots, \phi_k\}$ of the associated DMD operator. These capture the primary dynamic patterns in the time series. We represent $\mathcal{M}_k(\mathbf{X})$ as,

$$\mathcal{M}_k(\mathbf{X}) = \left[ \begin{array}{cccc} | & | & & | \\ \phi_1 & \phi_2 & \cdots & \phi_k \\ | & | & & | \end{array} \right]^\top \in \mathbb{R}^{k \times n}.$$

Selecting the number of retained modes $k$ involves a classical bias–variance trade-off: a smaller $k$ yields robustness to noise but may underrepresent the full system dynamics, whereas a larger $k$ enables more accurate reconstruction at the risk of overfitting. To formalize this trade-off, let $r = \mathrm{rank}(\boldsymbol{\Phi})$ denote the rank of $\boldsymbol{\Phi}$, i.e., the concatenation of all DMD eigenmodes. The approximation error can then be quantified using the Frobenius norm $\|\boldsymbol{\Phi} - \mathcal{M}_k(\mathbf{X})\|_F$. The exact expression for this error term is provided in Proposition 3.2.

**Proposition 3.2 (Eckart–Young–Mirsky theorem [14]).** *Let $\sigma_1 \geq \sigma_2 \geq \cdots \geq \sigma_r$ be the singular values of $\boldsymbol{\Phi}$. Then the DMD eigenmodes $\mathcal{M}_k(\mathbf{X})$ that uses the first $k$ modes satisfies,* $\|\boldsymbol{\Phi} - \mathcal{M}_k(\mathbf{X})\|_F = \left(\sigma_{k+1}^2 + \ldots + \sigma_r^2\right)^{1/2}$.

A common practical guideline is to select the smallest $k$ such that a prescribed proportion $\tau \in [0, 1]$ of the total energy is retained, i.e., $\sum_{j=1}^{k} \sigma_j^2 \geq \tau \sum_{j=1}^{r} \sigma_j^2$. In practice, a typical choice is $\tau = 0.95$.

## 3.2 Measuring the Similarity Between Time Series Using Their Respective DMD Modes

We are interested in quantifying the similarity of the underlying dynamics between $\mathbf{X}_i$ and $\widetilde{\mathbf{X}}_j$. Comparing their respective DMD eigenvectors provides a principled way to assess this similarity over time, as these eigenvectors capture and reveal the dominant dynamic patterns of the system. According to Equation 1, the dominant modes can be identified from the eigenvectors associated with the largest DMD eigenvalues. Consequently, we focus on comparing the similarity between the corresponding temporal mode subspaces, $\mathcal{M}_k(\mathbf{X})$ and $\mathcal{M}_k(\widetilde{\mathbf{X}})$. However, directly measuring the distance between these two subspaces is mathematically challenging, as their bases are not necessarily aligned. Although $\mathcal{M}_k(\mathbf{X})$ and $\mathcal{M}_k(\widetilde{\mathbf{X}})$ have the same dimensionality, their respective eigenvector subspaces are not necessarily aligned and may be expressed in different bases, making direct comparison nontrivial. To address this, we draw upon the concept of Grassmann manifolds from Information Geometry [2], which provides a natural and principled framework for comparing subspaces [27].

**Definition 3.3 (Grassmann manifold).** Let $V$ be an $n-$dimensional vector space The Grassmann manifold $\mathrm{Gr}(k, n)$ is the set of all $k-$dimensional subspaces of $V$, where $1 \leq k \leq n$. Mathematically, it can be expressed as $\mathrm{Gr}(k, n) = \{W \subseteq V : \dim(W) = k\}$.

The Riemannian distance between two subspaces on a Grassmann manifold is defined as the length of the shortest geodesic connecting them. This distance can be computed using the *principal angles* between the subspaces, which in our case correspond to the temporal modes.

**Definition 3.4 (Principal Angles Between Temporal Modes).** Let the columns of $\mathcal{M}_k(\mathbf{X})$ and $\mathcal{M}_k(\widetilde{\mathbf{X}})$ represent two linear subspaces $\mathbf{U}$ and $\mathbf{V}$, respectively. The principal angles $0 \leq \theta_1 \leq \cdots \leq \theta_r \leq \pi/2$ between the two subspaces are defined recursively as follows:

$$\cos\theta_k = \max_{u \in \mathbf{U}} \ \max_{v \in \mathbf{V}} u^\top v \qquad s.t. \ \begin{cases} u^\top u = v^\top v = 1 \\ u^\top u_i = v^\top v_i = 0, i = 1, \ldots, k-1 \end{cases} \tag{2}$$

The work of [8] has shown that the principal angles can be efficiently computed via the Singular Value Decomposition (SVD) of $\mathbf{Q}^\top \widetilde{\mathbf{Q}}$, where $\mathbf{Q}\mathbf{R}$ and $\widetilde{\mathbf{Q}}\widetilde{\mathbf{R}}$ denote the QR factorizations of $\mathcal{M}_k(\mathbf{X})$ and $\mathcal{M}_k(\widetilde{\mathbf{X}})$, respectively. The SVD of $\mathbf{Q}^\top \widetilde{\mathbf{Q}}$ can then be written as $\mathbf{Q}^\top \widetilde{\mathbf{Q}} = \mathbf{U}_{\mathrm{ang}}\mathbf{\Sigma}_{\mathrm{ang}}\mathbf{V}_{\mathrm{ang}}^\top$, where $\mathbf{U}_{\mathrm{ang}}$ and $\mathbf{V}_{\mathrm{ang}}$ are orthogonal matrices containing the left and right singular vectors, respectively, and $\mathbf{\Sigma}_{\mathrm{ang}}$ is a diagonal matrix whose entries correspond to the singular values associated with the principal angles between the two subspaces. If $s$ denotes the rank of $\mathbf{\Sigma}_{\mathrm{ang}}$, then the principal angles correspond to the arccosine of the first $s$ singular values of $\mathbf{\Sigma}_{\mathrm{ang}}$, i.e., $\mathbf{\Theta} = \mathrm{diag}\left(\cos^{-1}\sigma_1, \ldots, \cos^{-1}\sigma_s\right)$. Following [7], these principal angles can be used to define several Grassmannian metrics. One such metric is the *projection distance*, defined as the Frobenius norm of the matrix $\sin\mathbf{\Theta}$, i.e.,

$$d_{\mathrm{proj}}\left(\mathcal{M}_k(\mathbf{X}), \mathcal{M}_k(\widetilde{\mathbf{X}})\right) = \|\mathbf{\sin}(\mathbf{\Theta})\|_F = \left(\sum_{k=1}^{s} \sin^2\theta_k\right)^{1/2}. \tag{3}$$

The projection distance serves as a similarity metric between the temporal mode subspaces $\mathcal{M}_k(\mathbf{X})$ and $\mathcal{M}_k(\widetilde{\mathbf{X}})$. Smaller principal angles indicate that the subspaces are closer to each other, reflecting a higher degree of dynamical similarity. It is known that multiple geodesics can connect two points on the Grassmann manifold $\mathrm{Gr}(k, n)$. However, when all principal angles lie within the interval $[0, \pi/2]$, the corresponding geodesic is unique [53, 57].

## 3.3 Robustness of the DMD Mode Geodesic Distance

To verify that the proposed geodesic distance effectively captures mode collapse, it is essential to assess its stability under small perturbations in the system dynamics. Theorem 3.5 establishes an upper bound on this distance, showing that minor perturbations in the time series result in only small deviations in the subspace of DMD eigenvectors, thereby reinforcing the stability and reliability of the proposed metric.

**Theorem 3.5 (DMD reconstruction consistency).** *Let* $\mathbf{X} = [\mathbf{x}_1, \ldots, \mathbf{x}_\ell] \in \mathbb{R}^{n \times \ell}$ *and* $\widetilde{\mathbf{X}} = [\widetilde{\mathbf{x}}_1, \ldots, \widetilde{\mathbf{x}}_\ell] \in \mathbb{R}^{n \times \ell}$ *be two sequences of state snapshots. Assume that both* $\mathbf{X}$ *and* $\widetilde{\mathbf{X}}$ *admit*

*a DMD representation with the same initial condition* $\mathbf{x}_0 = \widetilde{\mathbf{x}}_0$. *Let* $\mathcal{M}_k(\mathbf{X}) \in \mathbb{C}^{n \times k}$ *and* $\mathcal{M}_k(\widetilde{\mathbf{X}}) \in \mathbb{C}^{n \times k}$ *denote the respective DMD mode matrices, associated with diagonal eigenvalue matrices* $\Lambda, \widetilde{\Lambda} \in \mathbb{C}^{k \times k}$. *Then, for all time steps* $t$, *the reconstructed states satisfy*

$$\mathbf{x}_t = \mathcal{M}_k(\mathbf{X}) \, \Lambda^t \, \mathcal{M}_k(\mathbf{X})^\dagger \mathbf{x}_0, \qquad \widetilde{\mathbf{x}}_t = \mathcal{M}_k(\widetilde{\mathbf{X}}) \, \widetilde{\Lambda}^t \, \mathcal{M}_k(\widetilde{\mathbf{X}})^\dagger \mathbf{x}_0.$$

Let $\mathbf{E}_t$ denote the difference in dynamics between $\mathbf{X}$ and $\widetilde{\mathbf{X}}$, i.e., $\forall t, \quad \mathbf{x}_t - \widetilde{\mathbf{x}}_t = \mathbf{E}_t \mathbf{x}_0$. Then, for all time steps $t$, the projection distance between the corresponding temporal mode subspaces satisfies $d_{\mathrm{proj}}\left(\mathcal{M}_k(\mathbf{X}), \mathcal{M}_k(\widetilde{\mathbf{X}})\right) \leq \frac{\|\mathbf{E}_t\|_F}{\delta_t}$, where $\delta_t$ denotes the spectral gap of $\Lambda^t$, and $\|\cdot\|_F$ represents the Frobenius norm. The proof of Theorem 3.5 is provided in Appendix C. This result demonstrates that small perturbations in the system dynamics lead to only minor variations in the subspace of DMD eigenvectors, ensuring that the geodesic distance remains a stable and reliable measure of dynamical similarity. In simpler terms, small differences in the time series translate into proportionally small differences in their DMD-based representations, making the proposed metric particularly well-suited for evaluating time-series generative models.

### 3.4 Measuring Mode Collapse For Time Series

To quantify mode collapse in generative models for time-series data, we propose a new approach based on Optimal Transport (OT) to assess the similarity and preservation of modes between real and generated samples. In this framework, DMD is first applied to both real and generated time series to extract their dominant modes, therefore capturing the key dynamical patterns underlying each dataset. For a given set of $L$ sampled batches of real time series $\mathcal{X} = \{\mathbf{X}_1, \mathbf{X}_2, \ldots, \mathbf{X}_L\}$ and generated time series $\widetilde{\mathcal{X}} = \{\widetilde{\mathbf{X}}_1, \widetilde{\mathbf{X}}_2, \ldots, \widetilde{\mathbf{X}}_L\}$, we compute the corresponding sets of DMD modes $\{\mathcal{M}_k(\mathbf{X}_i)\}_{i=1}^L$ and $\{\mathcal{M}_k(\widetilde{\mathbf{X}}_j)\}_{j=1}^L$, which encapsulate the dominant temporal dynamics of each sequence. We then construct a cost matrix $\mathbf{C}$, where each entry $\mathbf{C}_{ij}$ quantifies the dissimilarity between the mode subspaces $\mathcal{M}_k(\mathbf{X}_i)$ and $\mathcal{M}_k(\widetilde{\mathbf{X}}_j)$ using a principal-angle-based metric. The OT problem is subsequently solved to obtain the transport plan $\boldsymbol{\gamma}^\star$ that minimizes the total transportation cost, therefore identifying the optimal mapping between real and generated modes via the Wasserstein distance defined as follows:

$$d_{\mathrm{DMD}}(\mathcal{X}, \widetilde{\mathcal{X}}) = \mathbb{E}_{i,j}\left[W_p\left(\mathcal{M}_k(\mathbf{X}_i), \, \mathcal{M}_k(\widetilde{\mathbf{X}}_j)\right)\right] = \mathbb{E}_{i,j}\left[\min_{\boldsymbol{\gamma} \in \Pi} \langle \boldsymbol{\gamma}, \mathbf{C} \rangle_p\right], \qquad (4)$$

where $p$ denotes the order of the Wasserstein distance, $\Pi$ represents the set of all joint probability distributions, and $\langle \cdot, \cdot \rangle_p$ denotes the $p$-order cost-weighted inner product used to compute the transport cost. The resulting distance provides a robust measure of mode collapse: a smaller Wasserstein distance indicates better preservation of the original modes in the generated data, hence reflecting the effectiveness of the generative model in maintaining the intrinsic dynamical patterns of the time series. The geodesic $\boldsymbol{\gamma}$ in Equation 4 is defined formally in Theorem 3.6.

**Theorem 3.6 (DMD Mode Geodesic).** *Let* $\mathcal{M}_k(\mathbf{X}), \mathcal{M}_k(\widetilde{\mathbf{X}}) \in \mathbb{R}^{n \times k}$ *be matrices whose columns form orthonormal bases of two* $k$-*dimensional subspaces of* $\mathbb{R}^n$. *Let* $\boldsymbol{\Theta} = \mathrm{diag}(\theta_1, \theta_2, \ldots, \theta_k)$ *denote the diagonal matrix of principal angles between the subspaces spanned by* $\mathcal{M}_k(\mathbf{X})$ *and* $\mathcal{M}_k(\widetilde{\mathbf{X}})$. *Further, let* $\boldsymbol{\Delta} \in \mathbb{R}^{n \times k}$ *be an orthonormal matrix such that* $\mathcal{M}_k(\widetilde{\mathbf{X}}) = \mathcal{M}_k(\mathbf{X}) \cos(\boldsymbol{\Theta}) + \boldsymbol{\Delta} \sin(\boldsymbol{\Theta})$. *Then, the geodesic connecting* $\mathcal{M}_k(\mathbf{X})$ *and* $\mathcal{M}_k(\widetilde{\mathbf{X}})$ *on the Grassmann manifold* $\mathrm{Gr}(k, n)$ *is given by* $\boldsymbol{\gamma}(t) = \mathcal{M}_k(\mathbf{X}) \cos(t\boldsymbol{\Theta}) + \boldsymbol{\Delta} \sin(t\boldsymbol{\Theta})$, *for* $t \in [0, 1]$, *and the length of this geodesic corresponds exactly to the* projection distance:

$$\widetilde{d}_{\mathrm{proj}}\left(\mathcal{M}_k(\mathbf{X}), \mathcal{M}_k(\widetilde{\mathbf{X}})\right) = \left(\sum_{i=1}^k \theta_i^2\right)^{1/2}. \qquad (5)$$

Theorem 3.6 characterizes the geodesic path between two sets of temporal modes in time series data, showing that the transformation between these modes can be expressed precisely through trigonometric combinations of the principal angles (see proof in Appendix B). The distance $\widetilde{d}_{\mathrm{proj}}$ in Equation 5 is positively correlated with $d_{\mathrm{proj}}$ in Equation 3, since all principal angles lie within the interval $[0, \pi/2]$. This approach provides a quantitative and interpretable framework for evaluating the performance of generative models on time series data. We approximate the metric in Equation 4

**Algorithm 1 Computation of DMD-GEN Metric**

---

**Input:** Number of batches $B$

**Initialize:** $d_{\text{DMD}} \leftarrow 0$

**foreach** $l = 1, \ldots, B$ **do**

    Sample a batch of real time series $\mathcal{X}$ and generated time series $\widetilde{\mathcal{X}}$.

    **foreach** $\mathbf{X}_i \in \mathcal{X}$ **do**

        **foreach** $\widetilde{\mathbf{X}}_j \in \widetilde{\mathcal{X}}$ **do**

            Step 1. Extract temporal modes $\mathcal{M}_k(\mathbf{X}_i)$ from $\mathbf{X}_i$.

            Step 2. Extract temporal modes $\mathcal{M}_k(\widetilde{\mathbf{X}}_j)$ from $\widetilde{\mathbf{X}}_j$.

            Step 3. Compute orthonormal bases via QR factorization:

            $\mathcal{M}_k(\mathbf{X}_i) = \mathbf{Q}_i \mathbf{R}_i, \quad \mathcal{M}_k(\widetilde{\mathbf{X}}_j) = \widetilde{\mathbf{Q}}_j \widetilde{\mathbf{R}}_j.$

            Step 4. Compute principal angles from the SVD:

            $\mathbf{Q}_i^\top \widetilde{\mathbf{Q}}_j = \mathbf{U}_{\text{ang}} \cos(\mathbf{\Theta}) \mathbf{V}_{\text{ang}}^\top,$

            where $\cos(\mathbf{\Theta}) = \text{diag}(\cos\theta_1, \ldots, \cos\theta_r)$ and the principal angles are given by $\theta_\ell = \cos^{-1}(\sigma_\ell)$,

            with $\{\sigma_\ell\}_{\ell=1}^r$ being the singular values of $\mathbf{Q}_i^\top \widetilde{\mathbf{Q}}_j$.

            Step 5. Compute dissimilarity entry:

            $\mathbf{C}_{ij} = \widetilde{d}_{\text{proj}}\left(\mathcal{M}_k(\mathbf{X}_i), \mathcal{M}_k(\widetilde{\mathbf{X}}_j)\right).$

        **end foreach**

    **end foreach**

    Update metric estimate:

    $d_{\text{DMD}} \leftarrow d_{\text{DMD}} + \frac{1}{B} \min_{\boldsymbol{\gamma} \in \Pi} \langle \boldsymbol{\gamma}, \mathbf{C} \rangle_p.$

**end foreach**

**Output:** $d_{\text{DMD}}$

---

using the law of large numbers, and the detailed computational procedure is outlined in Algorithm 1. The values of the optimal transport matrix $\boldsymbol{\gamma}^\star = \arg\min_{\boldsymbol{\gamma} \in \Pi} \langle \boldsymbol{\gamma}, \mathbf{C} \rangle_p$ in Equation 4 quantify the extent to which the modes of each training time series are preserved in the generated samples.

**Time Complexity.** We analyze the computational complexity of Algorithm 1. Let $B$ be the number of batches, $S_b$ be the batch size, $n$ be the data dimensionality, $m$ be the time series length, $k$ be the number of modes, and $\epsilon$ be the OT solver precision. We assume $n, m \geq k$. The total complexity is driven by the $B$ outer loop iterations. In each iteration, we compute an $S_b \times S_b$ cost matrix $\mathbf{C}$ and solve the Optimal Transport (OT) problem. Computing $\mathbf{C}$ requires $S_b^2$ pairwise comparisons. The cost for each pair $(\mathbf{X}_i, \widetilde{\mathbf{X}}_j)$ is dominated by the two DMD extractions (Steps 1-2), which, as detailed in Appendix A.3 (Algorithm 2), have a complexity of $T_{\text{DMD}} = \mathcal{O}(nmk + nk^2 + mk^2 + k^3)$. This cost is higher than the subsequent QR factorization (Step 3: $\mathcal{O}(nk^2)$) and geodesic computation (Step 4: $\mathcal{O}(k^2 n + k^3)$). After building the matrix, solving the OT problem with Sinkhorn's algorithm [44] costs $\mathcal{O}(S_b^2/\epsilon)$. Therefore, the cost per batch is $\mathcal{O}(S_b^2 \cdot T_{\text{DMD}} + S_b^2/\epsilon)$. The total complexity for $B$ batches is: $\mathcal{O}(B \cdot S_b^2 \cdot (nmk + nk^2 + mk^2 + k^3 + 1/\epsilon))$.

## 4 Experiments

We evaluate the diversity of generative models across one synthetic dataset and three real-world datasets. The statistics of each dataset as well as the baselines metrics can be found in Appendix D.

**Consistency of DMD-GEN with Established Metrics.** Table 1 demonstrates that DMD-GEN produces results consistent with established evaluation metrics such as the Predictive Score, Discriminative Score, and Context-FID across all datasets. In each case, the rankings induced by DMD-GEN align closely with those from other metrics, effectively distinguishing between generative models based on their performance. A key advantage of DMD-GEN, however, lies in its efficiency: unlike other metrics, it requires no additional training to evaluate the generated time series. This makes DMD-GEN computationally efficient while maintaining consistent and reliable assessments of generative model quality.

As part of our ablation study, we also evaluated generic metrics not originally designed for time-series data but potentially applicable, such as MTopDiv [5], which measures divergence between original and generated samples by comparing data manifolds approximated as point clouds. Applying MTopDiv

Table 1: Comparison of generative model performance across multiple time series datasets using four evaluation metrics. Highlighted values indicate the best performance for each dataset. All metrics, including DMD-GEN, consistently identify the same best performing model, demonstrating strong agreement among evaluation methods. The symbol '–' denotes cases where computation failed or was not applicable.

| Metric | Model | Sines | ETTh | Stock | Energy |
|---|---|---|---|---|---|
| Disc. Score | TimeGAN | $0.03_{\pm 0.01}$ | $\mathbf{0.20}_{\pm \mathbf{0.03}}$ | $\mathbf{0.08}_{\pm \mathbf{0.04}}$ | $\mathbf{0.27}_{\pm \mathbf{0.04}}$ |
| | TimeVAE | $0.33_{\pm 0.02}$ | $0.50_{\pm 0.00}$ | $0.50_{\pm 0.00}$ | $0.50_{\pm 0.00}$ |
| | DiffusionTS | $\mathbf{0.02}_{\pm \mathbf{0.01}}$ | $0.50_{\pm 0.00}$ | $0.50_{\pm 0.00}$ | $0.50_{\pm 0.00}$ |
| Pred. Score | TimeGAN | $0.09_{\pm 0.00}$ | $\mathbf{12.39}_{\pm \mathbf{0.00}}$ | $\mathbf{6.40}_{\pm \mathbf{0.30}}$ | $\mathbf{24.01}_{\pm \mathbf{0.00}}$ |
| | TimeVAE | $0.12_{\pm 0.00}$ | $13.05_{\pm 0.03}$ | $27.12_{\pm 0.57}$ | $24.61_{\pm 0.06}$ |
| | DiffusionTS | $0.09_{\pm 0.00}$ | $13.18_{\pm 0.01}$ | $17.78_{\pm 0.08}$ | $24.49_{\pm 0.07}$ |
| Context-FID | TimeGAN | $0.04_{\pm 0.01}$ | $\mathbf{0.40}_{\pm \mathbf{0.05}}$ | - | $\mathbf{11.92}_{\pm \mathbf{1.95}}$ |
| | TimeVAE | $5.01_{\pm 1.04}$ | $12.22_{\pm 1.15}$ | - | $135.27_{\pm 22.83}$ |
| | DiffusionTS | $\mathbf{0.01}_{\pm \mathbf{0.00}}$ | $11.65_{\pm 0.76}$ | - | $127.02_{\pm 13.68}$ |
| DMD-GEN | TimeGAN | $33.91_{\pm 1.75}$ | $\mathbf{20.96}_{\pm \mathbf{1.10}}$ | $\mathbf{0.73}_{\pm \mathbf{0.19}}$ | $\mathbf{44.57}_{\pm \mathbf{7.34}}$ |
| | TimeVAE | $31.65_{\pm 0.51}$ | $98.91_{\pm 0.68}$ | $4.02_{\pm 0.08}$ | $164.48_{\pm 0.44}$ |
| | DiffusionTS | $\mathbf{29.66}_{\pm \mathbf{0.34}}$ | $105.46_{\pm 0.82}$ | $13.62_{\pm 2.53}$ | $150.67_{\pm 0.97}$ |

to time series requires flattening the sequences into independent samples, thereby discarding the essential temporal structure and feature dependencies. Nonetheless, for completeness, we conducted experiments using MTopDiv on the *Energy*, *ETTh*, and *Sines* datasets to compare the performance of TimeVAE, TimeGAN, and DiffusionTS (see Appendix F). The results revealed substantial variability in standard deviations across models and datasets, making meaningful comparison difficult. This instability suggests that MTopDiv is not a reliable metric for evaluating multivariate time-series generative models, as it fails to capture the intrinsic temporal dependencies of the data.

**Evaluating Metric Robustness Under Controlled Mode Collapse.** To study how different evaluation metrics behave under varying degrees of mode collapse, we created a synthetic dataset that allows direct control over the collapse severity. Specifically, we generated $N = 1000$ time series, each drawn from one of two distinct generators, $\mathcal{G}_1$ and $\mathcal{G}_2$, defined in Appendix E. Time series produced by the same generator are considered to belong to the same *mode*. The generator is chosen using a Bernoulli random variable with parameter $\lambda \in [0, 1]$: $\mathcal{G}_1$ is selected if $\lambda < \lambda_{\text{ref}}$, and $\mathcal{G}_2$ otherwise. The reference value $\lambda_{\text{ref}} = 0.5$ corresponds to a balanced mixture, where both modes are equally likely (co-exist) and no collapse occurs. We denote the resulting dataset by $\mathcal{D}_N(\lambda)$.

For each metric $m$, the non-collapse case is given by $m(\mathcal{D}_N(\lambda_{\text{ref}}), \mathcal{D}_N(\lambda_{\text{ref}}))$, while the collapsed cases correspond to $m(\mathcal{D}_N(\lambda_{\text{ref}}), \mathcal{D}_N(\lambda))$ for $\lambda \neq \lambda_{\text{ref}}$. Since the metrics have different numerical ranges, we compare them using a normalized performance measure:

$$\text{Perf}(\lambda) = \frac{m(\mathcal{D}_N(\lambda_{\text{ref}}), \mathcal{D}_N(\lambda))}{m(\mathcal{D}_N(\lambda_{\text{ref}}), \mathcal{D}_N(\lambda_{\text{ref}}))} - 1.$$

Table 2 reports $\text{Perf}(\lambda)$ for several metrics across different mode collapse levels. We observe that the benchmark metrics exhibit large fluctuations in both magnitude and sign as $\lambda$ varies, indicating sensitivity to changes in mode balance. In contrast, DMD-GEN remains stable and robust, effectively detecting even minor collapses. Both DMD-GEN and Context-FID increase rapidly as $\lambda$ deviates from $\lambda_{\text{ref}}$, signaling their sensitivity to emerging imbalances. However, unlike metrics such as Context-FID that can produce arbitrarily large values as discrepancies grow, DMD-GEN is naturally bounded by the geometry of the Grassmann manifold: the principal angles that define its distance lie within $[0, \pi/2]$. This bounded structure prevents extreme variations, ensuring numerical stability and making DMD-GEN a reliable and computationally efficient metric for detecting mode collapse in practice. Figure 1 illustrates the evolution of each evaluation metric as the mode collapse severity $\lambda$ increases. While Context-FID and DMD-GEN both exhibit a clear, monotonic increase reflecting stronger detection of collapse, DMD-GEN remains numerically stable across the entire range due to its bounded geometric formulation. In contrast, the Discriminative and Predictive Scores fluctuate considerably and fail to provide consistent trends, highlighting their limited sensitivity to gradual mode imbalances. These results confirm that DMD-GEN offers both sensitivity and robustness in detecting varying degrees of mode collapse.

Table 2: Relative performance of evaluation metrics in detecting increasing levels of synthetic mode collapse. DMD-GEN and Context-FID demonstrate strong sensitivity to collapse while maintaining stable behavior across severity levels.

| Metric | Mode Collapse Severity ($\lambda$) | | | | | |
|---|---|---|---|---|---|---|
| | **10%** | **20%** | **30%** | **40%** | **60%** | **70%** |
| Discr. Score | +586.79% | +443.40% | +181.13% | -20.75% | -16.98% | +143.40% |
| Pred. Score | -0.54% | -0.71% | -0.83% | -0.40% | +0.35% | +0.54% |
| Context-FID | +36,796.45% | +18,394.64% | +8,210.25% | +1,874.76% | +1,855.58% | +7,019.51% |
| DMD-GEN | +681.03% | +477.76% | +312.22% | +115.02% | +114.92% | +314.18% |

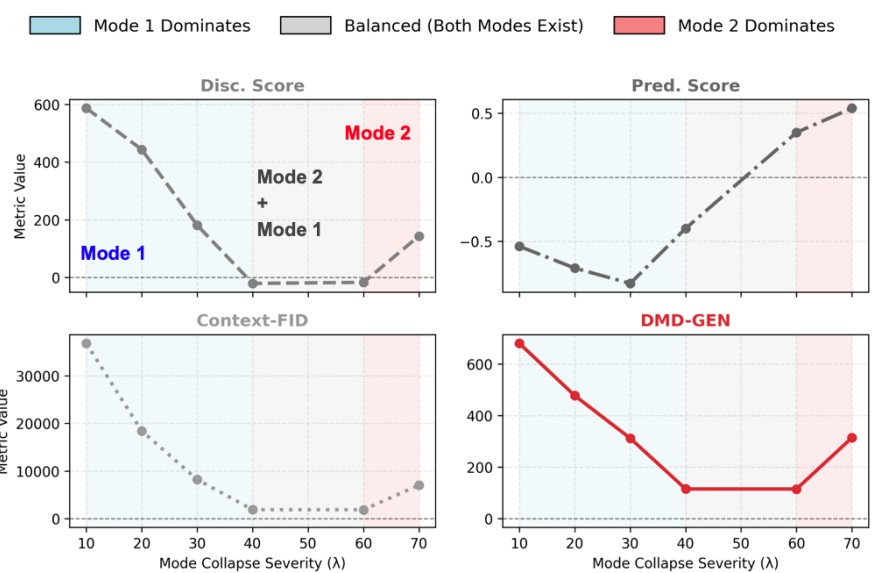

Figure 1: Comparison of metric sensitivity to varying mode collapse severity levels $\lambda$ in the synthetic dataset. Shaded regions indicate when one mode dominates (blue or red) versus when both modes coexist (gray). DMD-GEN and Context-FID show clear monotonic trends with $\lambda$, effectively distinguishing increasing collapse severity, whereas Predictive and Discriminative Scores fluctuate, indicating lower robustness to changes in mode balance.

**Assessing DMD-GEN on Bootstrapped Time Series.** Previous experiments focused primarily on deep learning-based generative models. To further validate the versatility of DMD-GEN, we evaluate it on time series generated through the classical non parametric Moving Block Bootstrap (MBB) method [16]. MBB preserves short-term temporal dependencies by resampling consecutive data blocks rather than individual points. To study how the choice of block size affects dynamic consistency, we apply MBB with three configurations: small blocks (introducing high randomness and weaker temporal coherence), medium blocks (partially retaining structure), and large blocks (preserving most temporal dependencies). We then compute the DMD-GEN distance between the original and bootstrapped time series to quantify how well dynamic patterns are maintained. As shown in Figure 2, increasing the block size consistently reduces DMD-GEN distance and stabilizes its variability, indicating that larger blocks better preserve the underlying temporal dynamics, whereas smaller blocks tend to distort them through excessive resampling noise.

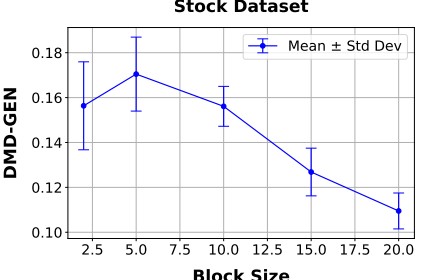

Figure 2: Mean DMD-GEN distance as a function of bootstrap block size for the Moving Block Bootstrap (MBB) experiment. Error bars denote standard deviations across trials. As block size increases, the DMD-GEN distance decreases and stabilizes, indicating improved preservation of temporal dynamics.

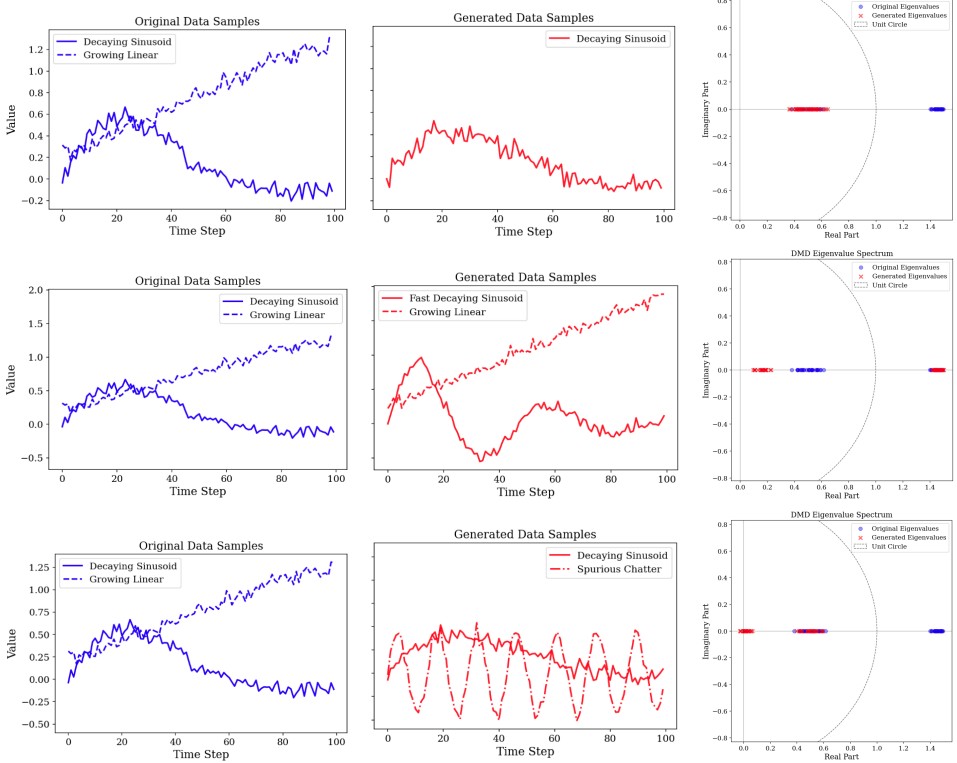

Figure 3: Visual interpretability of DMD-GEN across three representative cases: (top) Complete Mode Collapse, (middle) Dynamic Frequency Mismatch, and (bottom) Spurious Mode Injection. Each row illustrates how DMD-GEN responds to progressively more complex distortions in temporal structure, assigning higher distances to collapsed or over-generated modes and moderate, interpretable values to frequency mismatches. These examples demonstrate DMD-GEN's ability to quantify both the degree and the nature of dynamical discrepancies, with the following measured scores: Complete Mode Collapse = 0.7708, Dynamic Frequency Mismatch = 0.6833, and Spurious Mode Injection = 0.9114.

**Interpreting Mode Behavior Through the DMD Spectrum.** Figure 3 illustrates how the DMD spectrum reveals interpretable changes in system dynamics. In the top row, mode collapse concentrates dynamical activity into a few dominant components; the middle row shows frequency shifts indicating mild temporal distortion; and the bottom row displays additional spurious modes caused by injected noise. These spectral patterns provide a transparent view of how temporal structures deform across different dynamic perturbations.

## 5 Conclusions and Limitations

We introduced DMD-GEN, a new metric for evaluating generative models of time series and detecting mode collapse. By combining Dynamic Mode Decomposition with Optimal Transport, DMD-GEN provides a principled way to measure the similarity of temporal dynamics between real and generated data. Experiments show that DMD-GEN is more sensitive to mode collapse than existing metrics such as the Discriminative and Predictive Scores, while remaining consistent with their rankings. Unlike most existing metrics, DMD-GEN requires no additional training, making it computationally efficient and easy to apply. Its mode-based formulation also enhances interpretability by showing how key dynamical patterns are preserved or distorted in generated sequences. A current limitation is that DMD provides only a linear approximation of nonlinear dynamics: although Koopman theory allows such systems to be represented by an infinite-dimensional linear operator, practical approximations rely on a finite number of modes. Capturing stronger nonlinearities would therefore require expanding this set, increasing computational cost and reducing efficiency.

## Acknowledgments

The authors thank Kritaporn (Lune) Nitjaphanich for insightful discussions that helped shape this work.

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

# Supplementary Material: A Geometry-Aware Metric for Mode Collapse in Multivariate Time Series Generative Models

## A   Dynamical Mode Decomposition: Details and Proofs

### A.1   The Link Between the DMD operators in Continuous and Discrete Cases

Given a dynamical system $\dot{\mathbf{x}}(t) = \mathbf{f}(\mathbf{x}(t), t; \mu)$, we linearly approximate the dynamics using DMD using the operator $\mathcal{A} \in \mathbb{R}^{n \times n}$, i.e.

$$\forall t, \ \ \dot{\mathbf{x}}(t) = \mathcal{A}\mathbf{x}.$$

Discretizing time into intervals of $\Delta t$ and capturing snapshots accordingly, we establish the relationship between consecutive time steps in the following equation:

$$\forall k, \quad \mathbf{x}_{k+1} = \mathbf{x}_k + \mathcal{A}\mathbf{x}_k \Delta t = (I + \Delta t \mathcal{A})\, \mathbf{x}_k. \tag{6}$$

For a time-step $\Delta t$ that is sufficiently small, we can employ the first-order Taylor expansion of the matrix $exp(\Delta t \mathcal{A})$, expressed as:

$$exp(\Delta t \mathcal{A}) \approx I + \Delta t \mathcal{A} \tag{7}$$

Therefore, from Equations 6 and 7, we conclude that:

$$\forall k, \quad \mathbf{x}_{k+1} \approx exp(\Delta t \mathcal{A})\mathbf{x}_k.$$

Thus,

$$\mathbf{A}^\star \approx exp(\Delta t \mathcal{A}).$$

### A.2   Feasible Spectral Decomposition of the DMD Operator using Dimensionality Reduction

Algorithm 2 presents the steps to compute the eigenvectors and eigenvalues of the DMD operator $\mathbf{A}^\star$ using Singular Value Decomposition (SVD) for dimensionality reduction.

### A.3   DMD expansion

We prove the closed formula $\forall k, \quad \mathbf{x_k} = \sum_{j=1}^{r} \phi_j \lambda_j^k b_j = \mathbf{\Phi \Lambda^k b}$, using recursion.

For $k = 0$, we have,

$$\begin{aligned}
\mathbf{x_0} &= \mathbf{I x_0} \\
&= \mathbf{\Phi \Phi^\dagger x_0} \\
&= \mathbf{\Phi b} \\
&= \mathbf{\Phi \Lambda^0 b}
\end{aligned}$$

Let's now consider the equation hold for $k = 0, \ldots, m$, we have,

$$\begin{aligned}
\mathbf{x_{k+1}} &= \mathbf{A^\star x_k} \\
&= \mathbf{A^\star \Phi \Lambda^k b} \\
&= \mathbf{\Phi \Lambda \Lambda^k b} \\
&= \mathbf{\Phi \Lambda^{k+1} b}.
\end{aligned}$$

Therefore, the equality holds for all $k \in \mathbb{N}$.

---

**Algorithm 2** Dynamic Mode Decomposition

1. From collected snapshots of the system, build a pair of data matrices $(\mathbf{X}, \mathbf{X}')$.

$$\mathbf{X} = \begin{bmatrix} | & | & & | \\ \mathbf{x}_0 & \mathbf{x}_1 & \cdots & \mathbf{x}_{m-1} \\ | & | & & | \end{bmatrix}, \mathbf{X}' = \begin{bmatrix} | & | & & | \\ \mathbf{x}_1 & \mathbf{x}_2 & \cdots & \mathbf{x}_m \\ | & | & & | \end{bmatrix}$$

   The closed formula of optimal DMD operator is

$$\mathbf{A}^{\star} = \mathbf{X}'\mathbf{X}^{\dagger}$$

2. Compute the compact singular value decomposition (SVD) of $\mathbf{X}$:

$$\mathbf{X} \approx \mathbf{U}\mathbf{\Sigma}\mathbf{V}^{\dagger}$$

   where $U \in \mathbb{C}^{n \times r}, \Sigma \in \mathbb{C}^{r \times r}, V \in \mathbb{C}^{m \times r}$ and $r \leq min(m, n)$ is the rank of $\mathbf{X}$. Therefore,

$$\mathbf{A}^{\star} = \mathbf{X}'\mathbf{V}\mathbf{\Sigma}^{-1}\mathbf{U}^{\dagger}$$

3. Define a matrix

$$\tilde{\mathbf{A}} = \mathbf{U}^{\dagger}\mathbf{A}^{\star}\mathbf{U} = \mathbf{U}^{\dagger}\mathbf{X}'\mathbf{V}\mathbf{\Sigma}^{-1},$$

   since $U$ is a unitary matrix.

   $\tilde{\mathbf{A}} \in \mathbb{R}^{r \times r}$ defines a low-dimensional linear model of the dynamical system on proper orthogonal decomposition (POD) coordinates.

4. Compute the eigen-decomposition of $\tilde{\mathbf{A}}$:

$$\tilde{\mathbf{A}}\mathbf{W} = \mathbf{W}\mathbf{\Lambda},$$

   where columns of $\mathbf{W} \in \mathbb{R}^{r \times r}$ are eigenvectors and $\mathbf{\Lambda} = diag(\lambda_1, \ldots, \lambda_r) \in \mathbb{R}^{r \times r}$ is a diagonal matrix containing the corresponding eigenvalues.

5. Return DMD modes $\mathbf{\Phi}$:

$$\mathbf{\Phi} = \mathbf{X}'\mathbf{V}\mathbf{\Sigma}^{-1}\mathbf{W}.$$

   Each column of $\mathbf{\Phi}$ is an eigenvector of $\mathbf{A}$ meaning a DMD mode $\phi_k$ corresponding to eigenvalue $\lambda_k$

---

## B  Proof of Theorem 3.6 - DMD Mode Geodesic

**Theorem 3.6 [DMD Mode Geodesic]**

Let $\mathcal{M}_k(\mathbf{X}), \mathcal{M}_k(\widetilde{\mathbf{X}}) \in \mathbb{R}^{n \times k}$ be matrices whose columns form orthonormal bases of two $k$-dimensional subspaces of $\mathbb{R}^n$. Let $\Theta = \mathrm{diag}(\theta_1, \theta_2, \ldots, \theta_k)$ be the diagonal matrix of principal angles between the subspaces spanned by $\mathcal{M}_k(\mathbf{X})$ and $\mathcal{M}_k(\widetilde{\mathbf{X}})$. Let $\Delta \in \mathbb{R}^{n \times k}$ be an orthonormal matrix such that

$$\mathcal{M}_k(\widetilde{\mathbf{X}}) = \mathcal{M}_k(\mathbf{X})\cos(\Theta) + \Delta\sin(\Theta).$$

Then, the geodesic linking $\mathcal{M}_k(\mathbf{X})$ and $\mathcal{M}_k(\widetilde{\mathbf{X}})$ on the Grassmann manifold $\mathrm{Gr}(k, n)$ is given by

$$\gamma(t) = \mathcal{M}_k(\mathbf{X})\cos(t\Theta) + \Delta\sin(t\Theta), \quad \text{for } t \in [0, 1],$$

and the length of this geodesic corresponds exactly to the *projection distance* defined by

$$\widetilde{d}_{\mathrm{proj}}(\mathcal{M}_k(\mathbf{X}), \mathcal{M}_k(\widetilde{\mathbf{X}})) = \left(\sum_{i=1}^{k} \theta_i^2\right)^{1/2}.$$

*Proof.* **Preliminaries and Definitions**

1. **Grassmann Manifold** $\mathrm{Gr}(k, n)$: The set of all $k$-dimensional linear subspaces of $\mathbb{R}^n$.

2. **Orthonormal Bases**: For a $k$-dimensional subspace $\mathcal{S} \subset \mathbb{R}^n$, an orthonormal basis is represented by an $n \times k$ matrix $Q$ with columns satisfying $Q^{\top}Q = I_k$, where $I_k$ is the $k \times k$ identity matrix.

3. **Principal Angles and Vectors**: Given two subspaces $\mathcal{S}_1$ and $\mathcal{S}_2$ with orthonormal bases $Q_1$ and $Q_2$, the principal angles $0 \leq \theta_1 \leq \theta_2 \leq \cdots \leq \theta_k \leq \frac{\pi}{2}$ between them are defined recursively by

$$\cos(\theta_i) = \max_{\substack{\mathbf{u} \in \mathcal{S}_1 \\ \|\mathbf{u}\|=1}} \max_{\substack{\mathbf{v} \in \mathcal{S}_2 \\ \|\mathbf{v}\|=1}} \mathbf{u}^\top \mathbf{v}, \quad \text{subject to } \mathbf{u}^\top \mathbf{u}_j = 0, \ \mathbf{v}^\top \mathbf{v}_j = 0, \ j = 1, \ldots, i - 1. \quad (8)$$

4. **Projection Distance**: The projection distance between $\mathcal{S}_1$ and $\mathcal{S}_2$ is defined as

$$\widetilde{d}_{\text{proj}}(\mathcal{S}_1, \mathcal{S}_2) = \left( \sum_{i=1}^{k} \theta_i^2 \right)^{1/2}. \quad (9)$$

## 1. Computation of the Principal Angles

Let $Q_1 = \mathcal{M}_k(\mathbf{X})$ and $Q_2 = \mathcal{M}_k(\widetilde{\mathbf{X}})$. Both $Q_1$ and $Q_2$ are $n \times k$ matrices with orthonormal columns.

We construct the matrix $C$ as follows:

$$C = Q_1^\top Q_2 \in \mathbb{R}^{k \times k}. \quad (10)$$

Since $Q_1^\top Q_1 = I_k$ and $Q_2^\top Q_2 = I_k$, $C$ captures the pairwise inner products between the basis vectors of $Q_1$ and $Q_2$.

We then perform the Singular Value Decomposition (SVD) of $C$:

$$C = U \Sigma V^\top, \quad (11)$$

where

- $U, V \in \mathbb{R}^{k \times k}$ are orthogonal matrices, i.e., $U^\top U = V^\top V = I_k$.

- $\Sigma = \text{diag}(\sigma_1, \sigma_2, \ldots, \sigma_k)$ with $\sigma_i \geq 0$.

The singular values $\sigma_i$ of $C$ are the cosines of the principal angles between the subspaces:

$$\sigma_i = \cos(\theta_i), \quad \theta_i \in [0, \pi/2], \quad i = 1, \ldots, k. \quad (12)$$

This result stems from the fact that the SVD aligns the basis vectors of $U$ and $V$ to maximize the projections in the directions of the principal angles, which correspond to the largest cosines.

Since principal angles $\theta_i$ are defined in the range $[0, \pi/2]$, their cosines naturally lie in $[0, 1]$, matching the range of the singular values of $C$. Thus, the singular values encode the geometric relationship between the subspaces $U$ and $V$ in terms of the principal angles. This connection is fundamental to Grassmannian geometry, as it allows the distances and alignments between subspaces to be analyzed using the principal angles and their cosines.

## 2. Construction of Orthonormal Bases Aligned with Principal Directions

Define new orthonormal bases:
$$A = Q_1 U, \quad B = Q_2 V.$$

**Verification of Orthonormality:**

$$A^\top A = (Q_1 U)^\top (Q_1 U) = U^\top Q_1^\top Q_1 U = U^\top I_k U = U^\top U = I_k,$$
$$B^\top B = (Q_2 V)^\top (Q_2 V) = V^\top Q_2^\top Q_2 V = V^\top I_k V = V^\top V = I_k.$$

We then compute $A^\top B$:

$$A^\top B = (Q_1 U)^\top (Q_2 V) = U^\top Q_1^\top Q_2 V = U^\top C V = U^\top (U \Sigma V^\top) V$$
$$= U^\top U \Sigma V^\top V = I_k \Sigma I_k = \Sigma.$$

Thus, $A^\top B = \Sigma = \mathrm{diag}(\cos(\theta_1), \ldots, \cos(\theta_k))$.

## 3. Decomposition of $B$ in Terms of $A$ and $\Delta$

We aim to express $B$ as a linear combination of $A$ and another orthonormal matrix $\Delta$ that is orthogonal to $A$.

Let us define $\Delta$:

$$\Delta = (B - A\cos(\Theta))\sin(\Theta)^{-1}, \tag{13}$$

where $\cos(\Theta) = \Sigma$ and $\sin(\Theta) = \mathrm{diag}(\sin(\theta_1), \ldots, \sin(\theta_k))$, and $\sin(\Theta)^{-1}$ denotes the diagonal matrix with entries $\sin(\theta_i)^{-1}$.

**Verification that $\Delta$ is Orthogonal to $A$:**

$$
\begin{aligned}
A^\top \Delta &= A^\top (B - A\cos(\Theta))\sin(\Theta)^{-1} \\
&= (A^\top B - A^\top A\cos(\Theta))\sin(\Theta)^{-1} \\
&= (\Sigma - I_k \cos(\Theta))\sin(\Theta)^{-1} \\
&= (\cos(\Theta) - \cos(\Theta))\sin(\Theta)^{-1} = 0.
\end{aligned}
$$

**Verification that $\Delta$ is Orthonormal:**

First, we compute $\Delta^\top \Delta$:

$$
\begin{aligned}
\Delta^\top \Delta &= \left((B - A\cos(\Theta))\sin(\Theta)^{-1}\right)^\top \left((B - A\cos(\Theta))\sin(\Theta)^{-1}\right) \\
&= \sin(\Theta)^{-1}(B - A\cos(\Theta))^\top (B - A\cos(\Theta))\sin(\Theta)^{-1}.
\end{aligned}
$$

We compute the inner term:

$$
\begin{aligned}
(B - A\cos(\Theta))^\top (B - A\cos(\Theta)) &= (B^\top - \cos(\Theta)A^\top)(B - A\cos(\Theta)) \\
&= B^\top B - B^\top A\cos(\Theta) - \cos(\Theta)A^\top B \\
&\quad + \cos(\Theta)A^\top A\cos(\Theta).
\end{aligned}
$$

Since $A^\top A = I_k$, $B^\top B = I_k$, and $A^\top B = \Sigma = \cos(\Theta) = \cos(\Theta)^\top = (A^\top B)^\top = B^\top A$:

$$
\begin{aligned}
(B - A\cos(\Theta))^\top (B - A\cos(\Theta)) &= I_k - \cos(\Theta)^\top \cos(\Theta) - \cos(\Theta)^\top \cos(\Theta) \\
&\quad + \cos(\Theta)(I_k)\cos(\Theta) \\
&= I_k - \cos^2(\Theta) - \cos^2(\Theta) + \cos^2(\Theta) \\
&= I_k - \cos^2(\Theta) \\
&= \sin^2(\Theta).
\end{aligned}
$$

Since: $\sin^2(\Theta) + \cos^2(\Theta) = I_k$.

Thus,

$$\Delta^\top \Delta = \sin(\Theta)^{-1}\sin^2(\Theta)\sin(\Theta)^{-1} = (I_k).(I_k) = I_k.$$

Therefore, $\Delta$ is orthonormal.

**Expressing $B$ in Terms of $A$ and $\Delta$:**

Using Equation (13), we have:

$$B = A\cos(\Theta) + \Delta\sin(\Theta). \tag{14}$$

## 4. Define the Geodesic Path

On the Grassmann manifold, the geodesic $\gamma(t)$ from $A$ to $B$ is given by:

$$\gamma(t) = A\cos(t\Theta) + \Delta\sin(t\Theta), \quad t \in [0, 1]. \tag{15}$$

**Verification of Endpoints**:

At $t = 0$:

$$\gamma(0) = A\cos(0 \cdot \Theta) + \Delta\sin(0 \cdot \Theta) = AI_k + \Delta \cdot 0 = A.$$

At $t = 1$:

$$\gamma(1) = A\cos(\Theta) + \Delta\sin(\Theta) = B. \tag{16}$$

Thus, $\gamma(t)$ is a continuous path on $\mathrm{Gr}(k, n)$ connecting $A$ and $B$.

**Relate Back to Original Bases**:

Recall that $A = Q_1 U = \mathcal{M}_k(\mathbf{X})U$ and $B = Q_2 V = \mathcal{M}_k(\widetilde{\mathbf{X}})V$.

Since $U$ and $V$ are orthogonal matrices, the subspaces spanned by $Q_1$ and $A$, and by $Q_2$ and $B$, are identical. Therefore, we can express the geodesic in terms of $\mathcal{M}_k(\mathbf{X})$ and $\Delta$.

**Expressing the Geodesic in Original Terms**:

Let us redefine $\Delta$ accordingly to absorb $U$ and $V$, so that we can write:

$$\gamma(t) = \mathcal{M}_k(\mathbf{X})\cos(t\Theta) + \Delta\sin(t\Theta).$$

## 5. Compute the Length of the Geodesic

The length $L$ of the geodesic $\gamma(t)$ is given by:

$$L = \int_0^1 \|\dot{\gamma}(t)\|_F \; dt, \tag{17}$$

where $\|\cdot\|_F$ denotes the Frobenius norm.

**Compute the Derivative $\dot{\gamma}(t)$**:

Since $\gamma(t) = \mathcal{M}_k(\mathbf{X})\cos(t\Theta) + \Delta\sin(t\Theta)$, we have:

$$\dot{\gamma}(t) = -\mathcal{M}_k(\mathbf{X})\Theta\sin(t\Theta) + \Delta\Theta\cos(t\Theta),$$

where we used the fact that the derivative of $\cos(t\Theta)$ with respect to $t$ is $-\Theta\sin(t\Theta)$, and similarly for $\sin(t\Theta)$.

**Compute the Squared Norm $\|\dot{\gamma}(t)\|_F^2$**:

$$
\begin{aligned}
\|\dot{\gamma}(t)\|_F^2 &= \mathrm{Tr}\left(\dot{\gamma}(t)^\top \dot{\gamma}(t)\right) \\
&= \mathrm{Tr}\left((-\mathcal{M}_k(\mathbf{X})\Theta\sin(t\Theta) + \Delta\Theta\cos(t\Theta))^\top (-\mathcal{M}_k(\mathbf{X})\Theta\sin(t\Theta) + \Delta\Theta\cos(t\Theta))\right) \\
&= \mathrm{Tr}\left(\Theta^2\left(\sin^2(t\Theta)\mathcal{M}_k(\mathbf{X})^\top\mathcal{M}_k(\mathbf{X}) + \cos^2(t\Theta)\Delta^\top\Delta - \sin(t\Theta)\cos(t\Theta)\left(\mathcal{M}_k(\mathbf{X})^\top\Delta - \Delta^\top\mathcal{M}_k(\mathbf{X})\right)\right)\right).
\end{aligned}
$$

Since $\mathcal{M}_k(\mathbf{X})^\top\mathcal{M}_k(\mathbf{X}) = I_k$, $\Delta^\top\Delta = I_k$, and $\mathcal{M}_k(\mathbf{X})^\top\Delta = 0$, the cross terms vanish, and we have:

$$
\begin{aligned}
\|\dot{\gamma}(t)\|_F^2 &= \mathrm{Tr}\left(\Theta^2\left(\sin^2(t\Theta)I_k + \cos^2(t\Theta)I_k\right)\right) \\
&= \mathrm{Tr}\left(\Theta^2 I_k\right) \\
&= \sum_{i=1}^k \theta_i^2.
\end{aligned}
$$

**Compute the Length $L$:**

Since $\|\dot{\gamma}(t)\|_F$ is constant with respect to $t$, we have:

$$L = \int_0^1 \|\dot{\gamma}(t)\|_F \; dt = \|\dot{\gamma}(t)\|_F \int_0^1 dt$$

$$= \left( \sum_{i=1}^k \theta_i^2 \right)^{1/2} \cdot 1$$

$$= \left( \sum_{i=1}^k \theta_i^2 \right)^{1/2}.$$

## 6. Length Equals the Projection Distance

Comparing the computed length $L$ with the projection distance defined in Equation (9), we find:

$$L = \widetilde{d}_{\text{proj}}(\mathcal{M}_k(\mathbf{X}), \mathcal{M}_k(\widetilde{\mathbf{X}})) = \left( \sum_{i=1}^k \theta_i^2 \right)^{1/2}. \tag{18}$$

On the Grassmann manifold, the geodesic distance between two subspaces is given by the length of the shortest path connecting them. This distance is intrinsically linked to the principal angles between the subspaces. The projection distance quantifies the separation between subspaces in terms of these principal angles.

By computing the squared norm of the derivative of the geodesic, we find that it equals the sum of the squares of the principal angles, which is the squared projection distance. Since the derivative's norm is constant, the total length of the geodesic over the interval $t \in [0, 1]$ is precisely the projection distance.

Therefore, the length of the geodesic $\gamma(t)$ connecting $\mathcal{M}_k(\mathbf{X})$ and $\mathcal{M}_k(\widetilde{\mathbf{X}})$ on the Grassmann manifold equals the projection distance between these two subspaces.

This completes the proof of Theorem 3.6.

$\square$

## C  Proof of Theorem 3.5 - Metric Robustness

**Theorem 3.5** Let $\mathbf{X} = [\mathbf{x}_1, \ldots, \mathbf{x}_\ell] \in \mathbb{R}^{\ell \times n}$ and $\widetilde{\mathbf{X}} = [\widetilde{\mathbf{x}}_1, \ldots, \widetilde{\mathbf{x}}_\ell] \in \mathbb{R}^{\ell \times n}$ be two sequences of state snapshots. Suppose that both $\mathbf{X}$ and $\widetilde{\mathbf{X}}$ with $\mathcal{M}_k(\mathbf{X}) \in \mathbb{R}^{n \times k}$ and $\mathcal{M}_k(\widetilde{\mathbf{X}}) \in \mathbb{R}^{n \times k}$ as the respective DMD eigenvectors, $\Lambda$ and $\widetilde{\Lambda}$ as the respective DMD eigenvectors , and admit a DMD form with the same initial condition $\mathbf{x}_0 = \widetilde{\mathbf{x}}_0$, i.e.,

$$\forall t, \; \widetilde{\mathbf{x}}_t = \mathcal{M}_k(\mathbf{X}) \Lambda^t \mathcal{M}_k(\mathbf{X})^\dagger \mathbf{x}_0, \qquad \widetilde{\mathbf{x}}_t = \mathcal{M}_k(\widetilde{\mathbf{X}}) \widetilde{\Lambda}^t \mathcal{M}_k(\widetilde{\mathbf{X}})^\dagger \mathbf{x}_0.$$

Let $E_t$ be the difference in dynamics between $\mathbf{X}$ and $\widetilde{\mathbf{X}}$, i.e., $\forall t, \; \mathbf{x}_t - \widetilde{\mathbf{x}}_t = E_t x_0$. We have, $\forall t, \; d_{\text{proj}}(\mathcal{M}_k(\mathbf{X}), \mathcal{M}_k(\widetilde{\mathbf{X}})) \leq \frac{\|E_t\|_F}{\delta_t}$ where $\delta_t$ is the spectral gap of $\Lambda^t$, and $\|\cdot\|_F$ is the Frobenius Norm.

*Proof.* Let $X = [x_1, \ldots, x_\ell]$, $\quad \widetilde{X} = [\widetilde{x}_1, \ldots, \widetilde{x}_\ell] \in \mathbb{R}^{n \times \ell}$, and denote their $k$-dominant DMD bases by $Q := M_k(X) \in \mathbb{R}^{n \times k}, \widetilde{Q} := M_k(\widetilde{X}) \in \mathbb{R}^{n \times k}$. We define the time-$t$ linear propagators that generate the snapshots

$$A_t := Q\Lambda^t Q^\dagger, \qquad \widetilde{A}_t := \widetilde{Q}\widetilde{\Lambda}^t \widetilde{Q}^\dagger, \quad \text{so that} \quad x_t = A_t x_0, \; \widetilde{x}_t = \widetilde{A}_t x_0.$$

Because $x_0$ is arbitrary, we identify the perturbation matrix $A_t - \widetilde{A}_t = E_t$.

Wedin's theorem for diagonalizable matrices states that if $E$ perturbs a matrix $A$ whose spectrum splits into two clusters separated by a gap $\delta$, then $\|\sin\Theta\|_F \leq \frac{\|E\|_F}{\delta}$, where $\Theta$ collects the principal angles between the invariant subspaces associated with the chosen spectral clusters.

Applying Wedin with $A = A_t$, $E = E_t$, and the dominant invariant subspace $\mathrm{span}(Q)$, the gap is $\delta = \delta_t$. The left-hand side is exactly the Grassmann projection distance $d_{\mathrm{proj}}(Q, \widetilde{Q}) = \|\sin\Theta\|_F$. Hence $d_{\mathrm{proj}}(Q, \widetilde{Q}) \leq \frac{\|E_t\|_F}{\delta_t}$ which is the desired bound. $\qquad\square$

# D  Datasets and Implementation Details

## D.1  Basic Statistics on the Datasets

**Sine waves.** We generated a synthetic dataset consisting of two sets of sine waves to represent a bimodal distributed data. The data were generated using the following formula:

$$y(t) = A \cdot \sin(2\pi f t + \phi), \tag{19}$$

where $A$ is the amplitude, $f$ is the frequency, $t$ is the time variable and $\phi$ is the phase angle of the sine wave. Each mode consists of 2000 samples with phases being randomly chosen between 0 and $2\pi$. For all the samples, the duration is 2 seconds and the sample rate is 12, making the length of each sequence be 24. $A = 0.5$ and $f = 1$ Hz for the first mode, and $A = 5$ and $f = 0.5$ Hz for the second mode.

**Stock price.** To test our framework on a complex multimodal dataset, we used Google stocks data from 2004 to 2019, which was used in [60]. The data consists of 6 features which are daily open, high, low, close, adjusted close, and volume. The time series were then cut into sequences with length 24, following the setup in the work done by [60].

**Energy.** We conducted experiments on UCI's air quality dataset [55] consisting of hourly averaged responses from an array of 5 metal oxide chemical sensors embedded in an Air Quality Chemical Multisensor Device in an Italian city. Data was recorded from March 2004 to February 2005 and consists of 28 features. Unlike the previous datasets, this one has an unimodal distribution. The data is cut into several sequences of length 7.

**Electricity Transformer Temperature and humidity (ETTh).** The ETTh dataset focuses on temperature and humidity data from electricity transformers [62]. It includes 2 years of data at an hourly granularity, providing detailed temporal information about transformer conditions.

Table 3 provides an overview of the datasets used in our experiments, including Sine, Stock, Energy, and ETTh. These datasets vary in both the number of samples and feature dimensions, offering a diverse evaluation setting for generative models.

Table 3: Statistics of the four datasets used in our experiments.

| Dataset | Sine | Stock | Energy | ETTh |
|---|---|---|---|---|
| #Samples | 10,000 | 3,773 | 19,711 | 17,420 |
| Dimension | 5 | 6 | 28 | 8 |

## D.2  Baseline Metrics

We compared our proposed metric DMD-GEN with well-established time series evaluation metrics. Specifically, this comparison includes three key metrics:

**Predictive Score.** [60] The predictive score evaluates how well a generative model captures the temporal dynamics of the original data. It involves training a model on the generated data and assessing its performance on a real dataset. A lower predictive score indicates that the generated data contains patterns that are more representative of the temporal patterns found in the original data.

**Discriminative Score.** [60] The discriminative score measures the similarity between real and generated time series data by training a binary classifier to distinguish between them.

**Contextual Frechet Inception Distance (context-FID).** [22] Context-FID is an adaptation of the Frechet Inception Distance (FID), a metric used to assess the quality of images created by a generative model [19]. For time series, context-FID measures the similarity between the real and generated data distributions by computing the Frechet distance between feature representations extracted from a time series feature encoder.

### D.3  Implementation Details

The experiments were conducted on an NVIDIA A100 GPU. We utilized the pyDMD package [2] in Python to compute the DMD eigenvalues and eigenvectors. For generating synthetic time series, we used the original settings and the official implementation of DiffusionTS[3], TimeGAN[4] and TimeVAE[5].

## E  Synthetic Generators

To evaluate the ability of DMD-GEN to detect Mode Collapse, we generate synthetic time series using two parametric functions, denoted $\mathcal{G}_1$ and $\mathcal{G}_2$. These generators produce diverse temporal patterns by incorporating nonlinear transformations and oscillatory components. Each function is parameterized by randomly sampled variables from a uniform distribution, ensuring variability across generated samples. Below, we give the expressions of these generators,

$$\mathcal{G}_1 = \left\{ (t,x) \mapsto \frac{a}{\cosh(x+b+3)} \times \cos\left((c+2.3)\cdot t\right) \mid x \in [-5,5],\, t \in [0,4\pi],\, (a,b,c) \sim \mathcal{U} \right\},$$

$$\mathcal{G}_2 = \left\{ (t,x) \mapsto \frac{2+a}{\cosh(x)} \times \tanh(x) \times \sin\left((2.8+b)\cdot t\right) \mid x \in [-5,5],\, t \in [0,4\pi],\, (a,b,c) \sim \mathcal{U} \right\},$$

where $\mathcal{U}$ denotes the uniform distribution over $[0,1]$. Each time series is discretized to a length of $T = 129$ and a dimensionality of $d = 65$. Figure 4 illustrates examples of time series generated using $\mathcal{G}_1$ and $\mathcal{G}_2$. Generator $\mathcal{G}_1$ produces smooth, localized wave patterns with oscillations that gradually decay in space, resulting in broader and less frequent peaks over time. In contrast, $\mathcal{G}_2$ generates sharper, more structured wave patterns with higher frequency oscillations

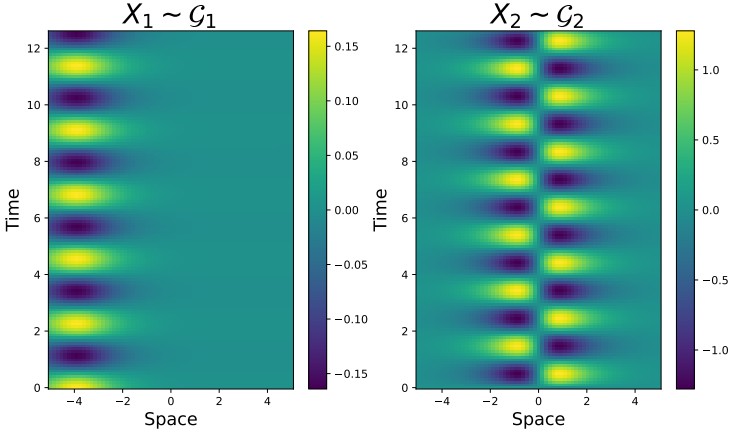

Figure 4: Examples of time series generated using the generators $\mathcal{G}_1$ and $\mathcal{G}_2$.

## F  Comparaison of Generative Models Using MTopDiv Metric

As part of our ablation study, we evaluated the MTopDiv metric, originally designed for general generative models, on time series data (Table 4). The results show high variability in standard

[2]https://pydmd.github.io/PyDMD/
[3]https://github.com/Y-debug-sys/Diffusion-TS
[4]https://github.com/Y-debug-sys/Diffusion-TS
[5]https://github.com/zzw-zwzhang/TimeGAN-pytorch

deviations, limiting meaningful comparisons and suggesting that MTopDiv is not well-suited for evaluating time series generative models.

| Dataset | TimeVAE | TimeGAN | DiffusionTS |
|---------|---------|---------|-------------|
| Energy  | 424.34 ±19.75 | 467.35 ±49.10 | 402.42 ±34.53± |
| ETTh    | 116.23 ±4.96± | 130.85 ±8.80  | 116.51 ±5.52 |
| Sines   | 7.72 ±0.18    | 4.92±0.32     | 4.53 ±0.13   |

Table 4: Comparison of different models on various datasets.

## G  Evolution of the DMD eigenvalues During Training

In Figures 5, and 6, we plot the imaginary and real parts of the DMD eigenvalues of a 500 sample original and generated time series for datasets ETTh and Sines.

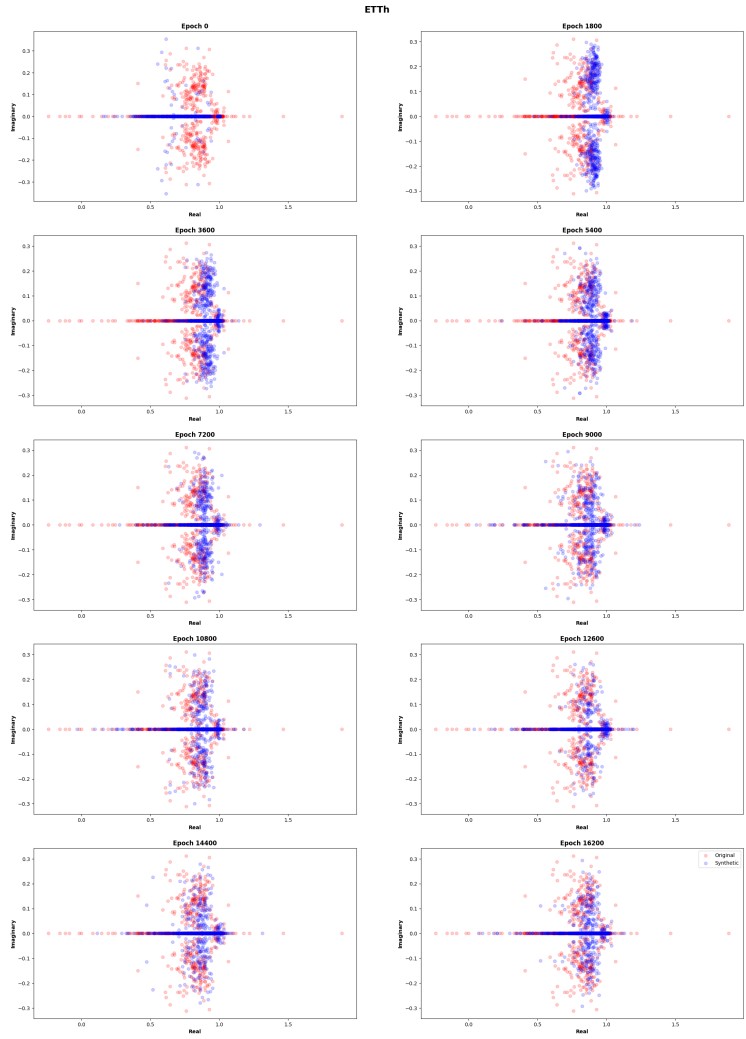

Figure 5: Comparison of DMD Eigenvalues between Original and Generated Time Series for DiffusionTS through Epochs on the dataset ETTh.

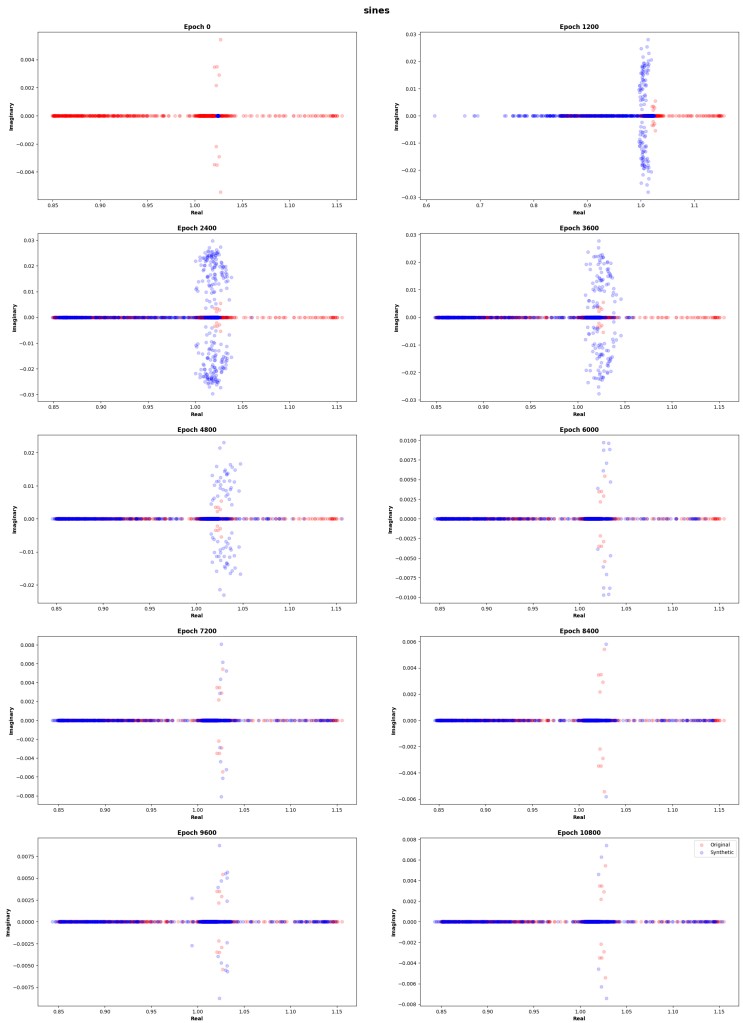

Figure 6: Comparison of DMD Eigenvalues between Original and Generated Time Series for DiffusionTS through Epochs on the dataset Sines.

# H  Broader Impact

This work aims to advance research in machine learning, particularly in the evaluation of generative models for time series data. Our goal is to improve the reliability and interpretability of such models, promoting the development of more transparent and trustworthy generative systems. We encourage responsible use of these methods and careful consideration of ethical implications in applied domains.

