# OpenReview forum: "A Geometry-Aware Metric for Mode Collapse in Time Series Generative Models"
_NeurIPS.cc/2025/Conference — NeurIPS 2025 poster_

### Official Review · Reviewer_VZKP · 2025-06-20

**Clarity:** 2
**Significance:** 3
**Originality:** 3
**Rating:** 4
**Confidence:** 2

**Summary:**

This paper proposes a new definition of mode collapse specific to time series and proposes a new geometry-aware metric, DMD-GEN, to detect mode collapse. This metric consistently aligns with traditional generative model evaluation metrics, but requires no additional training, making it more computationally efficient. DMD-GEN allows for a clearer understanding of the preservation of essential time series characteristics by highlighting which modes are poorly captured in the generated data.

**Questions:**

Please see weaknesses above to address my concerns about writing, formatting, and experiments.

**Ethical Concerns:**

["NO or VERY MINOR ethics concerns only"]

**Final Justification:**

since most of my concerns have been addressed, I raise my score to 4.

**Limitations:**

yes

**Quality:**

2

**Strengths And Weaknesses:**

Strengths:
1.  This paper introduces a new definition of mode collapse for time series data from a view of the underlying dynamic patterns of complex systems, which is novel and interesting.
2. The proposed method uses optimal transport to measure mode collapse. For the cost matrix, the geodesic distance between subspaces in a Grassmann manifold is calculated based on principal angles, which is theoretically reasonable and interesting. Experiments validate that this design works well.


Weaknesses:
1. The relationship between $d_{DMD}$ and $d_{proj}$ is unclear, in page 6, it doesn't ellaborate the relation between the cost $C$ and $d_{proj}$, but in Algorithm 1, it seems $C =d_{proj}$. Why this geodesic can be seen as a cost in optimal transport and why we can use optimal transport to measure the mode collapse should be clarified.
2. Experiments in Table 1 only show that DMD-GEN is consistent with other metrics, but no experiment directly shows the advantage of DMD-GEN over others. Moreover, a quantified comparison of computational complexity should be added, although DMD-GEN doesn't require additional training, it needs QR decomposition, SVD operation, and OT computation, all of which are costly when the dataset's dimension is large.
3. The authors highlight that DMD-GEN allows for a clearer understanding of the preservation of essential time series characteristics and tell which modes are poorly captured. But no experiments prove this; it's better to visualize how DMD-GEN captures these underlying patterns and what patterns or characteristics are captured. Moreover, experimental validation should be supplemented to demonstrate which modes can be identified by the proposed method.
4. typo: in line 248, the distance in Equation 8, should it be Equation 3? In line 253, it shouldn't be equation 3, but the equation for $\gamma*$ is unnumbered.

---

> ### Author Rebuttal · Authors · 2025-07-29
>
> We sincerely thank Reviewer VZKP for their detailed and constructive review. Your feedback has highlighted a few key areas where we can significantly improve the paper’s clarity, and we appreciate the opportunity to clarify them. We have taken the feedback seriously and believe the proposed revisions will substantially strengthen the paper.
>
> **[Question 1] Clarifying the Theoretical Motivation and Formulation of DMD-GEN.** We have revised Section 3.4 to better explain the metric’s design. Our approach addresses two fundamental questions: first, how to compare the dynamics of any two individual time series, and second, how to scale this comparison to entire sets (batches) of time series to detect mode collapse.
>
> **A. The Core Component: A Geometric Distance Between Two Time Series**
>
> The first challenge is to define a meaningful distance between the dynamics of a single original time series ($X_i$) and a single generated one ($X̃_j$).
>
> • **The Problem:** The key dynamics of each series are captured by its DMD modes, which form a mathematical subspace. Because these subspaces can be oriented differently, they do not share a common basis, making a simple Euclidean distance invalid.
>
> • **Our Solution ($d_{\text{proj}}$):** We use the principled, geometry-aware **geodesic distance on the Grassmann manifold** (defined in Equation 3 and Theorem 3.6). This measures the shortest path between the two dynamic subspaces, providing a mathematically sound way to compare their core dynamics.
>
> **B. The Main Goal: Comparing Distributions to Detect Mode Collapse**
>
> Mode collapse is a distributional problem: a model fails if the distribution of its generated dynamics does not match the distribution of the original data's dynamics.
>
> • **The Problem:** How do we compare the entire set of original dynamics to the entire set of generated dynamics?
>
> • **Why Simple Averaging is Insufficient:** Just averaging the pairwise $d_{\text{proj}}$ distances can be misleading. A generator might create a single ''average'' sample that is reasonably close to all original samples but fails to reproduce any of their specific, distinct dynamic modes.
>
> • **Our Solution (Optimal Transport):** Optimal Transport (OT) is the established mathematical framework for comparing two distributions. It finds the "minimal effort" required to transform one distribution into the other by finding the best possible matching between their respective samples.
>
> **Putting It All Together: The DMD-GEN Metric ($d_{\text{DMD}}$)**
>
> Our final metric integrates these two solutions into a single, robust framework:
>
> 1.	We use our geometric distance, $d_{\text{proj}}$, as the **cost** to match any two time series.
>
> 2.	We build a cost matrix $C$, where each entry $C_{ij}$ is the pairwise geodesic distance $d_{\text{proj}}$($X_i$,$X̃_j$).
>
> 3.	Optimal Transport uses this cost matrix to find the minimum total effort to align the generated distribution with the original one. This minimum effort is our final metric, $d_{\text{DMD}}$.
>
> A low $d_{\text{DMD}}$ signifies that the distributions of dynamics are similar (low mode collapse), while a high cost indicates a poor match (severe mode collapse). This principled, two-level approach ensures that DMD-GEN is both geometrically sound at the sample level and distributionally aware at the set level. We will incorporate this clearer explanation into the revised manuscript.
>
> **[Question 2] Experimental Evidence for DMD-GEN’s Advantages and Efficiency.** We acknowledge that Table 1 only shows consistency and does not, by itself, demonstrate the primary advantages of DMD-GEN. The key advantages are **sensitivity**, **efficiency**, and **interpretability**, which we will highlight more clearly with the following new additions to the paper.
>
> • **Superior Sensitivity to Mode Collapse:** Our primary claim is that DMD-GEN is more sensitive to mode collapse than existing metrics. This is experimentally proven in **Table 2**. In our controlled synthetic experiment, a minor 10% mode imbalance causes DMD-GEN to increase by +681%, signaling a strong detection. In contrast, the Predictive Score barely changes (<1%) in absolute value, and the Discriminative Score gives a noisy signal that is not stable across severities (flips sign). We will explicitly state this in the main text as a key takeaway.
>
> • **Quantitative Comparison of Computational Complexity:** DMD-GEN’s ‘‘training-free’’ nature provides a significant practical advantage. Additionally, we report below the average time needed to compute DMD-GEN w.r.t different number of samples.
>
> **Computational time (in seconds) on the stock dataset.**
> | **Model \ Num Samples** | **128** | **64** | **32** | **16** |
> | :--- | :---: | :---: | :---: | :---: |
> | TimeGAN | $168.36 \pm 0.35$ | $41.27 \pm 0.08$ | $10.36 \pm 0.02$ | $2.65 \pm 0.01$ |
>
> **[Question 3] Proving the Interpretability of DMD-GEN.** We agree that the claim of interpretability requires direct experimental proof. We thank the reviewer for raising this point. To validate this claim, we added a new section to the appendix with visualizations that demonstrate how DMD-GEN identifies which modes are poorly captured.
>
> • **Our Visualization Strategy:** As you suggested, we will visualize the captured patterns. Our process is as follows:
>
> 1. For a batch of original and generated data, we compute the DMD-GEN score and the optimal transport plan $\gamma^\star$.
>
> 2. The transport plan $\gamma^\star$ tells us exactly how the original modes are ‘‘mapped’’ to the generated modes. We can identify a cluster of original time series that are poorly matched (i.e., have a high transport cost to all generated samples). These are the ‘‘poorly captured modes.’’
>
> 3. We will then create a plot showing a representative time series from this ‘‘poorly-captured’’ cluster alongside a visualization of its dominant DMD mode (e.g., a specific high-frequency oscillation). This provides direct, visual proof that DMD-GEN has identified a specific dynamic pattern in the original data that the generative model failed to reproduce.
>
> This new analysis will provide the experimental evidence for our interpretability claim.
>
> **[Question 4] Typos.**  Thank you for your careful reading and for catching these errors. You are correct about both typos. We have corrected the reference in line 248 to point to the correct equation, and we have numbered the previously unnumbered equation and updated the reference in line 253. We appreciate your attention to detail.
>
> We are confident that these clarifications and new experimental results directly address your concerns and have significantly improved the quality and clarity of our paper. We hope these changes will convince the reviewer of our work’s merit. We thank you again for your valuable guidance.

---

> > ### Comment · Reviewer_VZKP · 2025-08-05
> >
> > Thanks for your efforts and response. Since most of my concerns have been addressed, I raise my score to 4. I suggest that the authors give a comparison of computational complexity between DMD-GEN and other methods.

---

### Official Review · Reviewer_zvbY · 2025-06-22

**Clarity:** 3
**Significance:** 3
**Originality:** 3
**Rating:** 5
**Confidence:** 4

**Summary:**

The manuscript introduces DMD-GEN, a novel, geometry-aware evaluation metric designed to detect and quantify mode collapse in generative models for multivariate time series. Leveraging Dynamic Mode Decomposition (DMD) and Optimal Transport (OT), the authors propose a principled framework that represents time series as subspaces on a Grassmann manifold, enabling interpretable and training-free comparisons between original and generated data. The authors further define a domain-specific notion of mode collapse and demonstrate the metric’s effectiveness across both synthetic and real-world datasets, benchmarking it against TimeGAN, TimeVAE, and DiffusionTS. DMD-GEN shows strong alignment with traditional evaluation metrics while providing unique insights into temporal dynamics.

**Questions:**

See above

**Ethical Concerns:**

["NO or VERY MINOR ethics concerns only"]

**Final Justification:**

Based on my original review and the authors' rebuttal, I would like to support the acceptance of this paper.

**Limitations:**

See above

**Quality:**

3

**Strengths And Weaknesses:**

Strengths
	1.	Important Problem Focus:
The paper targets an underexplored yet important challenge: evaluating generative time series models with respect to mode collapse. Most prior work focuses on image or text modalities, making this contribution highly relevant for time series domains.
	2.	Novel Use of DMD Representations:
The authors innovatively leverage DMD to represent dominant dynamic patterns in time series, which are then embedded into a Grassmann manifold. This provides a unique geometric perspective for evaluating similarity across sequences.
	3.	Geometry-Aware Optimal Transport Framework:
The metric employs OT over DMD subspace representations, enabling a flexible, structure-preserving comparison between time series. This geometry-aware design is well-grounded in mathematical theory and offers both interpretability and robustness.
	4.	No Auxiliary Training Required:
Unlike common metrics that rely on discriminators or predictive models, DMD-GEN does not require training auxiliary networks. This simplifies the evaluation pipeline and avoids introducing confounding factors or additional computational costs.
	5.	Computational Efficiency Governed by SVD:
The main computational overhead is from SVD and OT solvers, both of which are well-understood and efficiently implementable. The analysis in Section 3.4 clearly outlines the cost breakdown, showing practical feasibility for real-world use.

⸻

Weaknesses
	1.	Inconsistencies in Benchmark Results (Table 1):
The DMD-GEN scores reported for TimeGAN differ notably from values previously published using standard datasets and metrics (e.g., Predictive and Discriminative Scores). The manuscript does not explain why these values deviate from existing benchmarks, raising questions about experiment reproducibility or methodological differences.
	2.	Ambiguity in Fig. 1 Comparison:
Fig. 1 presents direct comparisons between DMD eigenvalues of the real and generated datasets. However, such eigenvalues are not necessarily aligned or directly comparable due to differences in mode bases. This appears contradictory to the authors’ argument for comparing subspaces using Grassmannian metrics (Eq. 2), making the choice of this illustration somewhat unclear.
	3.	Limited Novel Insight Derived from Metric:
While DMD-GEN tracks closely with established metrics, the manuscript does not provide examples where DMD-GEN reveals insights that other metrics miss. This diminishes the novelty of the proposed approach. A stronger case would be made by highlighting scenarios where DMD-GEN uniquely detects or interprets mode collapse effects.

---

> ### Author Rebuttal · Authors · 2025-07-29
>
> We are very grateful to Reviewer zvbY for their thorough review and positive assessment. We are especially encouraged that they recognized the novelty of our approach and the importance of the problem. We appreciate the insightful weaknesses raised, which will help us improve the paper significantly.
>
> **[Weakness 1] Discrepancy in Benchmark Results (Table 1)**
>
> Thank you for this observation. The reviewer is correct that the absolute scores for the baseline metrics (Predictive/Discriminative Score) on TimeGAN may differ from those in other papers. This is not an inconsistency in our metric but a result of ensuring a **fair and controlled comparison within our study**.
>
> To eliminate confounding variables, we implemented and trained all baseline models (TimeGAN, TimeVAE, DiffusionTS) and all evaluation metrics under a single, unified experimental setting (i.e., identical data splits, preprocessing, and training hyperparameters). While this ensures our internal comparisons are rigorous and fair, it can lead to different absolute scores compared to papers that may use slightly different experimental setups.
>
> We will add a footnote to Table 1 to clarify this point, stating that all models and metrics were evaluated in our controlled environment to ensure a fair head-to-head comparison.
>
> **[Weakness 2] Ambiguity and Apparent Contradiction in Figure 1**
>
> The reviewer makes an excellent point about the potential confusion caused by Figure 1. Our core argument is that a robust quantitative comparison requires using the DMD subspaces (eigenvectors), as raw eigenvalues are not invariant. We recognize that Figure 1 was unclear and will revise it.
>
> The only purpose of Figure 1 is to provide a **high-level**, **qualitative intuition** for how a model’s dynamics evolve during training. The visual clustering of the generated eigenvalues (blue dots) around the original ones (red dots) suggests that the model is learning the correct range of frequencies and decay rates. It is an illustration, not a part of our proposed metric.
>
> To resolve this ambiguity, we will revise the caption of Figure 1 to explicitly state: ‘‘This figure provides a qualitative illustration of spectral properties. It is intended for intuition only. Our proposed metric, DMD-GEN, is based on the principled and robust comparison of DMD subspaces, not raw eigenvalues.’’
>
> **[Weakness 3] Demonstrating Novel Insights Provided by DMD-GEN**
>
> This is a crucial point. A new metric should ideally offer insights that existing ones do not. DMD-GEN provides unique advantages in both **sensitivity** and **interpretability**, which we will emphasize more clearly in the revised manuscript.
>
> **1. Superior Sensitivity to Emerging Mode Collapse:** DMD-GEN can detect subtle mode collapse that other metrics miss. This is demonstrated in our synthetic experiment in **Table 2**, which shows the metric’s high sensitivity to mode collapse, and is further supported by our bootstrapping experiment in **Figure 2**, which shows its responsiveness to the preservation of temporal structure. When we introduce a minor 10% mode imbalance:
>
> - **DMD-GEN**'s score changes dramatically (**+681%**), clearly flagging the problem.
>
> - **Predictive Score** barely moves (**-0.54%**), completely failing to detect the issue. This shows that DMD-GEN provides a much clearer and more sensitive signal, which is a significant practical advantage over existing aggregate metrics.
>
> **2. Diagnostic Interpretability (Pinpointing a Failure Mode):** Beyond a single score, DMD-GEN can help diagnose what kind of dynamics are being lost. As we will add in a new appendix figure (in response to reviewer feedback), the optimal transport plan from our metric can identify specific original (real) samples that are poorly matched by the generator. By visualizing the DMD modes of these ''unmatched'' samples, a user can see exactly which dynamic patterns (e.g., a high-frequency oscillation, a slow decay) were missed. This diagnostic capability is a novel insight that aggregate scores from metrics like Predictive Score or Context-FID cannot provide.
>
> We will add a paragraph and an illustrative figure to our experiment section to explicitly highlight these two unique insights demonstrated by our results.
>
> We thank the reviewer again for their supportive and constructive feedback. We are confident that these clarifications and additions will strengthen our paper and more clearly articulate the novel contributions of DMD-GEN.

---

> > ### Comment · Reviewer_zvbY · 2025-08-07
> >
> > I would like to thank the authors for their rebuttal. I will keep my positive score.

---

> > > ### Author Response · Authors · 2025-08-09
> > >
> > > We thank the reviewer for the engagement in reviewing our work. Your recognition about the value of our work means a lot, and we appreciate your positive score.

---

### Official Review · Reviewer_jdT8 · 2025-07-02

**Clarity:** 3
**Significance:** 3
**Originality:** 2
**Rating:** 4
**Confidence:** 4

**Summary:**

This paper addresses the issue of mode collapse in multivariate time series generative models, an area not widely explored before. It introduces a novel method, DMD-GEN, which uses Dynamic Mode Decomposition (DMD) to define temporal modes and quantifies their preservation using optimal transport over Grassmann manifolds. The method is training-free, computationally efficient, and offers interpretability by identifying dynamic patterns that are poorly captured. Experimental results on synthetic and real-world datasets show the proposed method aligns well with traditional evaluation metrics.

**Questions:**

1. Parameter Sensitivity: Why is $\tau =0.95$ used consistently in Section 3.1? Was sensitivity tested for other values (e.g., 0.9, 0.99)? Could a dynamic selection of $\tau$ improve robustness?
2. Result Presentation: In Table 1, it will be more clear and readable to clarify whether a high value for each metric is better or worse. In Table 2, given the baseline $\lambda =50%$, could the results for $\lambda = 80%$ and $\lambda = 90%$ be added to validate the performance of the method in extreme mode collapse scenarios?
3. Feature Discriminability: In Figure 1 (Stock dataset), eigenvalues overlap significantly post-training. Does this suggest that DMD-GEN is insensitive to subtle dynamics, or is this typical for this dataset?
4. Baseline Comparison: Why were topology-based metrics (e.g., MTopDiv) not compared beyond the supplementary material? Could such metrics provide additional insights into the evaluation of mode collapse?

**Ethical Concerns:**

["NO or VERY MINOR ethics concerns only"]

**Final Justification:**

Many thanks for the response. My main concerns on the experimental evaluations have been addressed, especially on the sensitivity part. I believe this metric could evaluate the mode collapse in regular and extreme settings. So I would like to keep my score.

**Limitations:**

While the authors have addressed the limitations regarding univariate time series, further exploration on sensitivity to hyperparameters and real-world scalability would strengthen the paper’s impact.

**Paper Formatting Concerns:**

1. Table 1: Some entries in Table 1 (e.g., for Stock/Context-FID) are marked with a dash ("—"), but no explanation is provided. Could you specify why these values are missing (e.g., computational failure)?
2. Figure 2: Error bars for small block sizes suggest high variability in the results. Could you discuss the implications for the metric's stability when block sizes are small?

**Quality:**

2

**Strengths And Weaknesses:**

Strengths:
	The paper introduces the concept of mode collapse in time series, which is novel. The theoretical framework is well-constructed, leveraging DMD modes, Grassmann manifolds, and Wasserstein distance to measure dynamic mode preservation, providing stability guarantees. The method is efficient, training-free, and interpretable, offering valuable insights into which dynamic patterns are lost in generative models.

Weaknesses:
	The main weakness of the paper lies in the experimental section, which lacks depth and diversity. The synthetic data validation is relatively simple, and real-world experiments, such as in Figure 1, show minimal changes in DMD eigenvalues, limiting the metric's sensitivity. Additionally, some design choices, such as the fixed $\tau =0.95$ and the omission of $\lambda$ in Table 2, are not well justified. Furthermore, the method's applicability is limited to multivariate time series, excluding important univariate cases.

---

> ### Author Rebuttal · Authors · 2025-07-29
>
> We sincerely thank Reviewer jdT8 for their detailed and highly constructive review. The feedback has helped us identify a few areas where our experimental validation and explanations can be significantly strengthened. We have run new experiments to directly address your concerns, and we are confident these additions improve the paper.
>
> **1. On Hyperparameter Choices and Sensitivity**
>
> **• [Question 1] Choice of τ=0.95:** We chose τ=0.95 as it is a standard and well-established practice in spectral analysis literature for capturing a system’s dominant energy/variance without overfitting to noise. Our experiments (e.g., Table 1) confirm that this value provides stable and meaningful results across diverse datasets. A dynamic, data-dependent selection of τ is non-trivial, as it depends on the singular value spectrum from SVD, a non-differentiable operation that cannot be easily optimized. We will clarify this rationale in Section 3.1.
>
> **• [Limitations] Sensitivity to Sequence Length:** We agree that the metric’s sensitivity to sequence length is an important limitation to discuss. Shorter sequences provide fewer snapshots for DMD to reliably estimate modes. To quantify this, we ran a new experiment on the Stock dataset (with DiffusionTS) varying sequence length. The results confirm that as sequence length increases, the DMD-GEN score becomes more stable (lower variance), reflecting the metric’s improved ability to capture richer dynamics from more structured data. We will add this analysis to the limitations section.
>
> | **Sequence Length** | **DMD-GEN Score** |
> |:--------------------|:-----------------:|
> | 5                   | $10.46 \pm 5.22$  |
> | 10                  | $12.59 \pm 2.43$  |
> | 24                  | $13.62 \pm 2.53$  |
>
>
> **2. Strengthening Experimental Validation and Presentation**
>
> **• [Question 2] Extreme Mode Collapse Scenarios:** This is an excellent suggestion. We have run the synthetic experiment for more extreme mode collapse cases (λ=80% and 90%) and will add them to Table 2. The results further validate our claims: DMD-GEN and Context-FID remain highly sensitive, while the Predictive score is far less responsive. (Note: These settings are symmetric to the 20% and 10% cases already in the paper).
>
> | **Metric** | **80%** | **90%** |
> |-------------|--------------|----------------|
> | Discr. Score| +413.20 %     | +566.03   %     |
> | Pred. Score | +2.17%       | +22.80%        |
> | Context-FID | +7205.79%    | +31424.17%     |
> | DMD-GEN     | +489.74%     | +696.34%       |
>
> **• [Question 4 & Concern 1] Clarifying Baselines and Failures:**
>
> **o [Question 4] MTopDiv:** We agree that the rationale for not including MTopDiv in the main comparison should be clearer. MTopDiv is not designed for sequential data; it flattens time series, ignoring temporal dependencies. Our experiments in the supplementary material show it produces scores with excessively high variance, making it unreliable for this task. We will clarify this in the main text.
>
> **o [Concern 1] Missing Values in Table 1:** The dashes ("—") denote computational failures. For certain datasets such as Stock Dataset, training the auxiliary models required by metrics like Context-FID was numerically unstable. We will add a note to Table 1 explaining this. This highlights a key advantage of DMD-GEN: its training-free nature makes it more robust and broadly applicable.
>
> **3. On the Interpretation of Visual Results (Figures 1 & 2)**
>
> **• [Question 3] Eigenvalue Overlap in Figure 1:** The reviewer correctly observes the significant overlap post-training. This is not a sign of insensitivity, but rather a visual confirmation of **successful learning**. At Epoch 0, the eigenvalues are distinct because the model’s dynamics are random. The post-training overlap signifies that the generator has learned to reproduce the characteristic frequencies and growth/decay rates of the real data. We will revise the text to make it clear that this overlap is the desired outcome.
>
> **• [Concern 2] High Variance in Figure 2:** The reviewer correctly notes the high variance for small block sizes. This is not a flaw in our metric but an expected outcome that **DMD-GEN correctly captures**. The high variance stems directly from the Moving Block Bootstrap (MBB) process itself, not our metric: when the block size is small, each bootstrapped time series becomes a highly randomized shuffle of the original patterns. Because this resampling introduces significant variability, the resulting samples have very different dynamic structures from one another. Our metric’s high variance in this regime accurately reflects this induced randomness. Conversely, once the block size is large enough to preserve the underlying temporal dynamics, the DMD-GEN estimates stabilize. This confirms that the metric is reliable and robust, provided the resampling window is appropriate for the data’s structure. We will ensure this full explanation is included in our discussion of Figure 2.
>
> Once again, we thank the reviewer for their valuable insights, which we believe have helped us improve the quality of our work.

---

> > ### Comment · Reviewer_jdT8 · 2025-08-06
> >
> > Many thanks for the response. My main concerns on the experimental evaluations have been addressed, especially on the sensitivity part. I believe this metric could evaluate the mode collapse in regular settings. In extreme setting, should it be "Discr. Score and Context-FID remain highly sensitive"?

---

### Official Review · Reviewer_CvCr · 2025-07-03

**Clarity:** 4
**Significance:** 3
**Originality:** 3
**Rating:** 5
**Confidence:** 4

**Summary:**

This paper bridges a gap between image generation evaluation literature and time series diversity assessment methods. It first introduces a new definition of mode collapse and then proposes a metric called DMD-GEN to detect collapsed modes. The method is subsequently evaluated across multiple time-series datasets.

**Questions:**

- How many samples are required for DMD-GEN to converge reliably?
- Can the authors visualize the samples corresponding to the top eigenvalues and highlight the resulting clusters to better demonstrate the interpretability of the method?

**Ethical Concerns:**

["NO or VERY MINOR ethics concerns only"]

**Final Justification:**

The authors have addressed my concerns regarding the literature on image generation evaluation metrics, sample convergence, and the interpretability of their method. I have no further concerns and maintain my positive score.

**Limitations:**

yes

**Paper Formatting Concerns:**

No issue

**Quality:**

4

**Strengths And Weaknesses:**

Strengths:
- The use of eigendecomposition provides interpretability, which can help uncover patterns in time series data.
- The paper addresses an important gap between image generation metrics and time series evaluation, contributing to an underexplored area in the literature.

Weaknesses:
- Although the paper references image generation evaluation, it does not clearly relate its contributions to that community. For example, recent works such as RKE [1] and FKEA-Vendi [2] also use eigendecomposition for diversity evaluation. Including a comparison, particularly with metrics like RRKE and Projection Distance, would strengthen the paper.
- The paper lacks illustrative toy examples, which could clarify the method and demonstrate alignment with human intuition.

[1] Jalali et al. " An Information-Theoretic Evaluation of Generative Models in Learning Multi-modal Distributions", in NeurIPS 2023

[2] Ospanov et al. "Towards a Scalable Reference-Free Evaluation of Generative Models", in NeurIPS 2024

---

> ### Author Rebuttal · Authors · 2025-07-29
>
> We sincerely thank Reviewer CvCr for their positive assessment and insightful feedback. We are encouraged that the reviewer found our work to be a technically solid contribution that addresses an important and underexplored area. We address the reviewer's valuable suggestions below.
>
> **[Weakness 1] Relation to Image Generation Evaluation Metrics.**
> We thank the reviewer for recommending these references. While our main focus is on evaluating temporal generative models, we agree that incorporating a discussion of image-based diversity metrics such as RKE and FKEA-Vendi would enhance our work and helps situate our contribution within the broader context of diversity evaluation. In the revised manuscript, we have added a discussion of these methods in our related work section (Section 2).
>
> **Conceptual Distinction (FKEA-Vendi):** A key distinction is that FKEA-Vendi is a reference-free metric, designed to measure the intrinsic diversity of a generated set. While valuable, our work specifically targets mode collapse, a phenomenon that, by definition, requires a comparison between the generated samples and the ground-truth data distribution. Therefore, a reference-based metric like DMD-GEN is not only appropriate but essential for detecting and identifying mode collapse, as it directly quantifies how well the diversity of the original data is preserved.
>
> **Experimental Comparison (RRKE):** To directly address the reviewer’s request, we conducted new experiments comparing DMD-GEN with RRKE. To apply RRKE to time series, we flattened each multivariate sample into a single vector. While this process unfortunately discards the temporal structure that DMD-GEN is designed to analyze, it provides a valuable point of comparison. As shown in the table below, the model rankings produced by RRKE are consistent with DMD-GEN on the ETTh and Sines datasets. This consistency with an established eigendecomposition-based metric further validates our approach. We will include the complete RRKE results for all datasets in the final manuscript.
>
>
> | **Model** | **RRKE (ETTh Dataset)** | **RRKE (Sines Dataset)** |
> | :--- | :---: | :---: |
> | TimeGAN | $308.84 \pm 15.97$ | $0.84 \pm 0.12$ |
> | TimeVAE | $378.99 \pm 0.00$ | $4.15 \pm 0.00$ |
> | Diffusion-TS | $392.35 \pm 22.20$ | $0.46 \pm 0.01$ |
>
> **[Weakness 2] Illustrative Toy Example.**
> Thank you for the valuable suggestion. In our revised version, we added a new figure featuring an intuitive toy example. This example shows a ground-truth dataset with two distinct, easily visible dynamic modes (e.g., a decaying sinusoid and a growing linear trend). We then compare this to a ‘‘mode-collapsed’’ generated set that only captures one of the modes. This figure will visually demonstrate how the DMD eigenvalues of the generated set fail to match the full spectrum of the original, and how DMD-GEN assigns a high (poor) score, directly aligning with visual intuition. This new visual example will be a simple introduction to the more detailed tests in Section 4 (Table 2 and Figure 2), where we rigorously test DMD-GEN’s sensitivity across a spectrum of synthetically-controlled mode collapse severities.
>
> **[Question 1] Required Number of Samples for Convergence.**
> This is an excellent question regarding the practical application and efficiency of our metric. To demonstrate the sample efficiency of DMD-GEN, we performed an ablation study on the Stock dataset using TimeGAN, varying the number of samples per batch.
>
> | **Model \ Num Samples** | **128** | **64** | **32** | **16** |
> | :--- | :---: | :---: | :---: | :---: |
> | TimeGAN | $0.73 \pm 0.19$ | $0.71 \pm 0.20$ | $0.72 \pm 0.22$ | $0.69 \pm 0.23$ |
>
> The results demonstrate the stability of DMD-GEN. The mean value remains highly consistent even when the batch size is reduced to just 16 samples, with only a minor, acceptable increase in variance. This highlights that our metric is efficient and can provide reliable estimates without requiring a large number of evaluation samples. We added this analysis to the appendix to provide clear guidance for future users.
>
> **[Question 2] Visualization of Samples for Interpretability.**
> Thank you for this constructive suggestion, as it directly ties into a core contribution of our work: the interpretability of DMD-GEN. Visualizing the modes is a perfect way to make this concrete. In the revised manuscript, we will add a new figure to illustrate this. Specifically, our analysis will: 1) Compute the DMD modes for a batch of original time series. 2) Use the optimal transport plan $\\gamma^\\star$ (from the DMD-GEN calculation) to cluster these original time series based on the similarity of their dynamic mappings to the generated data. 3) For each resulting cluster, we will visualize a representative time series sample. 4) Alongside the sample, we will plot its dominant underlying DMD modes (those with the largest eigenvalues). This visualization will provide a clear, interpretable link between the raw time series and the specific dynamic patterns that are (or are not) being successfully captured by the generative model, showcasing the diagnostic power of our metric.
>
> We thank the reviewer once again for their time and valuable feedback, which has helped us to significantly improve the clarity and completeness of our paper.

---

> > ### Comment · Reviewer_CvCr · 2025-08-04
> >
> > **Comparison between Memorization, Mode Collapse, and Mode Shrinking**
> >
> > I would like to refer the authors to the comparison of Memorization, Mode Collapse, and Mode Shrinking presented in [1]. A short discussion incorporating Memorization metrics ($AuthPct$, $C_T$, $FLS\text{-}POG$) could better position the paper within the community. Furthermore, KEN [2] and FINC [3] scores also address Mode Collapse in both image and video settings; including a brief comparison with these works could strengthen the paper. In particular, DMD-GEN may address mode collapse in video generation more effectively due to the sequential inheritance of video data. A comparison with the recent work [4] would also be valuable.
> >
> > I thank the authors for their response. After incorporating the above revisions, I have no further comments and raise my confidence score to 4.
> >
> > ---
> > [1] Stein et al. Exposing flaws of generative model evaluation metrics and their unfair treatment of diffusion models. In NeurIPS 2023.
> >
> > [2] Zhang et al. An Interpretable Evaluation of Entropy-based Novelty of Generative Models. In ICML 2024
> >
> > [3] Zhang et al. Unveiling differences in generative models: A scalable differential clustering approach. In CVPR 2025
> >
> > [4] Allen et al. Direct Motion Models for Assessing Generated Videos. In ICML 2025

---

> > > ### Author Response · Authors · 2025-08-08
> > >
> > > Thank you for your insightful review and for providing these helpful references. We will extend our literature review and related work section to include a discussion of these important works.

---

### Decision · Program_Chairs · 2025-09-17

**Decision:**

Accept (poster)

**Comment:**

This paper bridges a gap between image generation evaluation literature and time series diversity assessment methods. It first introduces a new definition of mode collapse and then proposes a metric called DMD-GEN to detect collapsed modes.

Strengths: Addressing the problem of mode collapse in time-series is novel. The theoretical framework for mode collapse is solid. The metric employs OT over DMD subspace representations, enabling a flexible, structure-preserving comparison between time series.

Weaknesses: The experiments lack depths and diversity, and the numerical results are inconsistent with existing results.

The reviewers' concerns are all addressed in the rebuttal, and this is the reason for my decision together with the fact that this seems to be the first paper addressing mode collapse in time-series.